# Re-examining how complexin inhibits neurotransmitter release

**Thorsten Trimbuch[1][†], Junjie Xu[2][†], David Flaherty[2], Diana R Tomchick[2], Josep Rizo[2,3,4]\*, Christian Rosenmund[1]\***

[1]NeuroCure Cluster of Excellence, Neuroscience Research Center, Charité-Universitätsmedizin Berlin, Berlin, Germany; [2]Department of Biophysics, University of Texas Southwestern Medical Center, Dallas, United States; [3]Department of Biochemistry, University of Texas Southwestern Medical Center, Dallas, United States; [4]Department of Pharmacology, University of Texas Southwestern Medical Center, Dallas, United States

**Abstract** Complexins play activating and inhibitory functions in neurotransmitter release. The complexin accessory helix inhibits release and was proposed to insert into SNARE complexes to prevent their full assembly. This model was supported by 'superclamp' and 'poor-clamp' mutations that enhanced or decreased the complexin-I inhibitory activity in cell–cell fusion assays, and by the crystal structure of a superclamp mutant bound to a synaptobrevin-truncated SNARE complex. NMR studies now show that the complexin-I accessory helix does not insert into synaptobrevin-truncated SNARE complexes in solution, and electrophysiological data reveal that superclamp mutants have slightly stimulatory or no effects on neurotransmitter release, whereas a poor-clamp mutant inhibits release. Importantly, increasing or decreasing the negative charge of the complexin-I accessory helix inhibits or stimulates release, respectively. These results suggest a new model whereby the complexin accessory helix inhibits release through electrostatic (and perhaps steric) repulsion enabled by its location between the vesicle and plasma membranes.

**\*For correspondence:** jose@arnie.swmed.edu (JR); christian.rosenmund@charite.de (CR)

[†]These authors contributed equally to this work

**Reviewing editor**: Axel T Brunger, Stanford University, United States

## Introduction

Neurotransmitter release is crucial for interneuronal communication and is exquisitely regulated by a sophisticated protein machinery (*Sudhof, 2013*). Great advances have been made in elucidating the mechanism of release (*Brunger et al., 2009*; *Sorensen, 2009*; *Jahn and Fasshauer, 2012*; *Rizo and Sudhof, 2012*) and basic aspects of this process have been reconstituted with eight central components of the release machinery (*Ma et al., 2013*), leading to a model with defined roles for each component. In this model, the neuronal soluble N-ethylmaleimide-sensitive factor attachment protein receptors (SNAREs) synaptobrevin, syntaxin-1 and SNAP-25 form a tight four-helix bundle called the SNARE complex (*Sollner et al., 1993*; *Poirier et al., 1998*; *Sutton et al., 1998*) that brings the synaptic vesicle and plasma membranes together (*Hanson et al., 1997*) and is critical for membrane fusion; N-ethylmaleimide-sensitive factor (NSF) and soluble NSF attachment proteins (SNAPs) disassemble the SNARE complex (*Sollner et al., 1993*) to recycle the SNAREs for another round of fusion (*Mayer et al., 1996*), and may favor physiological membrane fusion by disassembling syntaxin-1-SNAP-25 complexes (*Ma et al., 2013*); Munc18-1 and Munc13s orchestrate SNARE complex assembly in an NSF/SNAP resistant manner (*Ma et al., 2013*), and may play a direct role in fusion (*Dulubova et al., 2007*; *Li et al., 2011b*); and synaptotagmin-1 acts as a $Ca^{2+}$ sensor (*Fernandez-Chacon et al., 2001*), likely by bridging the two membranes (*Arac et al., 2006*; *Xue et al., 2008a*).

Tight regulation of neurotransmitter release also depends critically on complexins, small soluble proteins that bind to the SNARE complex (*McMahon et al., 1995*) and play activating and inhibitory

**eLife digest** The instructions sent to, from and within the brain are rapidly transmitted along neurons in the form of electrical signals. These signals cannot pass across the small gaps—called synapses—that separate neighboring neurons. Instead, neurons release chemicals called neurotransmitters into the synapses, and these relay the signal to the next neuron.

The neurotransmitters are stored inside neurons in small bubbles called vesicles. To release these neurotransmitters into the synapse, the membrane that encloses the vesicle fuses with the membrane that surrounds the neuron. To fuse the membranes, proteins embedded in the vesicle membrane interact with similar proteins in the neuron membrane to form a structure called a SNARE complex. Additional proteins control membrane fusion to ensure that the signal is passed to the other neuron at the right time and with the appropriate efficiency.

Among these proteins are the complexins, which are often found attached to SNARE complexes. Although different parts of complexins can both help and hinder membrane fusion, a part known as an accessory helix is thought to have only one role—to stop the membranes from fusing together. Several models have been suggested for how the accessory helix interferes with fusion. However, after performing a range of analyses by diverse biophysical techniques, Trimbuch, Xu et al. suggest these models are unlikely to describe the process accurately.

Instead, Trimbuch, Xu et al. propose a new model based on the electrostatic properties of two molecules that are both negatively charged. An accessory helix taken from a fruit fly complexin was more negatively charged than a mammalian version, and experiments showed it was also better at preventing the release of neurotransmitters. It is thought that the negative charges on the helix hold the membranes apart because the helix is located between the membranes, which are also negatively charged. Consistent with this model, Trimbuch, Xu et al. showed that the membranes fused more easily when some of the negative charges on the accessory helix were replaced with positive charges. The next challenges are to test the model further with additional studies, and to explain how other proteins work with complexins to control neurotransmitter release.

functions. Absence of complexins leads to a severe impairment of $Ca^{2+}$-evoked exocytosis and to varied effects on spontaneous release ranging from small decreases to dramatic increases, depending on the system (*Reim et al., 2001*; *Huntwork and Littleton, 2007*; *Xue et al., 2008b*; *Maximov et al., 2009*; *Hobson et al., 2011*; *Martin et al., 2011*; *Yang et al., 2013*). These results likely arise from an interplay between stimulatory and inhibitory activities of different regions of complexins (*Xue et al., 2007*, *2009*; *Cho et al., 2010*; *Kaeser-Woo et al., 2012*). Complexin I (CpxI) is largely unstructured in solution (*Pabst et al., 2000*) but forms a central α-helix that binds to the SNARE complex and is preceded by an accessory helix (*Bracher et al., 2002*; *Chen et al., 2002*) (*Figure 1A*). The central helix is crucial for both the activating and inhibitory functions of complexins, while the accessory helix inhibits release (*Xue et al., 2007*; *Maximov et al., 2009*); the complexin N-terminus plays an activating function, releasing the inhibition by the accessory helix (*Xue et al., 2010*; *Yang et al., 2010*), and the C-terminal sequence has activating and inhibitory roles (*Kaeser-Woo et al., 2012*).

Cell–cell fusion assays and reconstitution studies also indicated dual roles for complexins that likely recapitulate to some extent their physiological functions (*Giraudo et al., 2006*; *Schaub et al., 2006*; *Yoon et al., 2008*; *Malsam et al., 2009*, *2012*; *Diao et al., 2012*), but in most cases these studies revealed only stimulatory or inhibitory roles. The activating function has been proposed to arise from stabilization of the SNARE complex by complexin binding (*Chen et al., 2002*), from interactions of the complexin N-terminus with the C-terminus of the SNARE complex (*Xue et al., 2010*), and from binding of the complexin C-terminal region to phospholipids (*Seiler et al., 2009*), but these models remain to be validated. The inhibitory activity of complexins attracted much attention because several studies suggested that complexins prevent exocytosis before $Ca^{2+}$ influx and synaptotagmin-1 releases the inhibition upon $Ca^{2+}$ binding by displacing complexins from the SNARE complex (*Giraudo et al., 2006*; *Schaub et al., 2006*; *Tang et al., 2006*; *Roggero et al., 2007*). Later analyses showed that CpxI is not fully displaced but there is competition between synaptotagmin-1 and part of CpxI for binding to the SNARE complex on membranes (*Dai et al., 2007*; *Xu et al., 2013*). Genetic interaction studies in hippocampal neurons showed that complexins regulate release similarly in the absence or presence

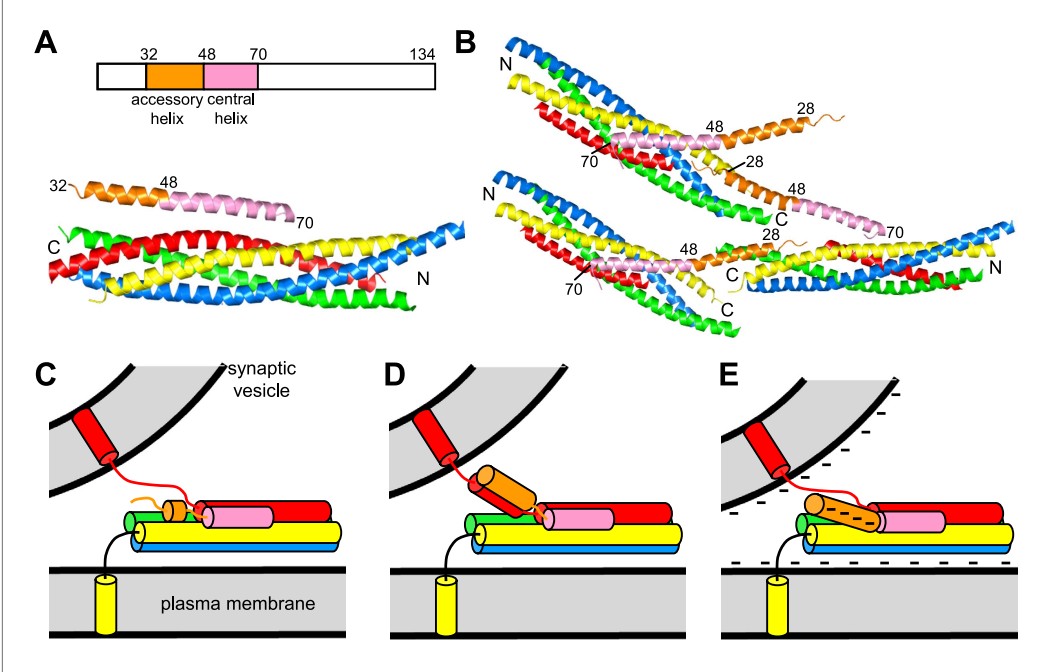

**Figure 1**. Models of complexin function. (**A**) Domain diagram of CpxI and ribbon diagram of the crystal structure of the CpxI(26-83)/SNARE complex (PDB code 1KIL) (**Chen et al., 2002**). Selected residue numbers are indicated above the ribbon diagram and on the CpxI(26–83) ribbon in the structure. Synaptobrevin is colored in red, syntaxin-1 in yellow, SNAP-25 in blue and green (N-terminal and C-terminal SNARE motifs, respectively), and CpxI(26-83) in orange (accessory helix) and pink (central helix). The same color code is used in all panels. N and C indicate the N- and C-termini of the SNARE motifs. (**B**) Ribbon diagram of the crystal structure of the complex between the CpxI(26-83) superclamp mutant and a synaptobrevin truncated SNARE complex (PDB code 3RK3) (**Kummel et al., 2011**). Three complexes are shown to illustrate the zigzag array present in the crystals. (**C**–**E**) Models for the inhibitory activity of the complexin accessory helix. In all models, the accessory helix is proposed to prevent C-terminal assembly of a partially assembled SNARE complex either by inserting into the complex (**C**), by binding to the synaptobrevin SNARE motif (**D**), or by electrostatic repulsion with both membranes (**E**).

The following figure supplements are available for figure 1:

**Figure supplement 1**. The interface between CpxI and the SNARE complex.

**Figure supplement 2**. High B-factors in the accessory helix in the crystal structure of the CpxI(26-83) superclamp mutant bound to SCΔ60.

of synaptotagmin-1, indicating that complexins function independently of synaptotagmin-1 (**Xue et al., 2007**, **2010**), but this finding does not exclude the notion that complexins and synaptotagmin-1 may cooperate in regulating release. For example, the dramatic increase of spontaneous release observed in complexin nulls in *Drosophila* requires synaptotagmin-1 (**Jorquera et al., 2012**), and absence of complexins in hippocampal neurons sensitizes release to loss of 50% of synaptotagmin-1 expression (**Xue et al., 2010**).

While it is clear that the complexin accessory helix inhibits release (**Xue et al., 2007**; **Maximov et al., 2009**), a satisfactory model for this activity has not emerged yet. We initially proposed that part of this helix might replace part of the synaptobrevin SNARE motif in trans partially assembled SNARE complexes, hindering C-terminal assembly of the complex (**Xue et al., 2007**; **Figure 1C**). A similar insertion model, but envisioning that the entire accessory helix replaces part of synaptobrevin, was proposed later and was supported by the enhanced inhibition in cell–cell fusion assays caused by replacing charged with hydrophobic residues in the accessory helix of CpxI ('superclamp mutants') and by the design of a 'poor-clamp' mutation (K26A) that impairs the inhibitory activity (**Giraudo et al., 2009**). The crystal structure of a fragment of a CpxI superclamp mutant (D27L, E34F, R37A) bound to

a SNARE complex with C-terminally truncated synaptobrevin suggested an alternative model whereby the central helix of one CpxI molecule binds to a SNARE complex and the accessory helix inserts into another SNARE complex, resulting in a zigzag array (*Kummel et al., 2011*; *Figure 1B*). However, formation of such a complex with wild type (WT) CpxI would be highly unfavorable thermodynamically because three charged residues would be placed into hydrophobic environments.

The study described here was designed to investigate how the complexin accessory helix inhibits neurotransmitter release, testing the insertion and zigzag models as well as additional models that emerged subsequently (*Figure 1D,E*). Using NMR spectroscopy and isothermal titration calorimetry (ITC), we show that the accessory helix of CpxI does not insert into synaptobrevin-truncated SNARE complexes in solution. Furthermore, in stark contrast with the cell–cell fusion data, rescue experiments in complexin I-III triple knockout (KO) neurons reveal that superclamp mutations in CpxI lead to slightly stimulatory or no effects on neurotransmitter release, while the poor-clamp K26A mutation impairs release. We also find that the accessory helix of complexin from *drosophila melanogaster* inhibits spontaneous release more strongly than the accessory helix of mammalian CpxI, which may arise from the more negatively charged nature of the former. Indeed, a mutation that increases the negative charge of the CpxI accessory helix inhibits release and a mutation that decreases the negative charge enhances release. These results suggest a model whereby the location of the negatively charged accessory helix between the synaptic vesicle and plasma membranes causes electrostatic and perhaps steric repulsion with the membranes, thus hindering membrane fusion and neurotransmitter release (*Figure 1E*).

## Results

### The accessory helix does not insert into synaptobrevin-truncated SNARE complexes: NMR analysis with $^2$H,$^{15}$N-labeled CpxI fragments

To analyze interactions between CpxI and soluble truncated SNARE complexes that might mimic trans SNARE complexes partially assembled between two membranes (e.g., *Figure 1C*, insertion model), we used $^1$H-$^{15}$N two-dimensional transverse relaxation optimized spectroscopy (TROSY) heteronuclear single quantum coherence (HSQC) spectra, which provide a powerful tool to study protein–protein interactions. These NMR spectra can be viewed as protein fingerprints with one cross-peak for each non-proline residue in a $^{15}$N-labeled protein, and the positions and line widths of the cross-peaks are very sensitive to perturbations caused by binding to an unlabeled protein (*Rizo et al., 2012*). Flexible and unstructured regions exhibit sharp cross-peaks with poor dispersion whereas structured regions have broader, well-dispersed cross-peaks, as exemplified by $^1$H-$^{15}$N TROSY-HSQC spectra of a uniformly $^2$H,$^{15}$N-labeled CpxI fragment spanning the accessory and central helices [CpxI(26-83)]. As described previously (*Chen et al., 2002*), the $^1$H-$^{15}$N TROSY-HSQC spectrum of this fragment exhibits sharp cross-peaks and poor dispersion (*Figure 2A*) because, although partially helical, the fragment is very flexible. Upon binding to a minimal SNARE complex containing the SNAREs motifs of synaptobrevin, syntaxin-1 and SNAP-25 (below referred to as SNARE complex or SC), the $^1$H-$^{15}$N TROSY-HSQC spectrum of CpxI(26-83) reveals strong broadening and a dramatic increase in dispersion for cross-peaks from the central helix (*Figure 2B*, red contours) due to stable packing of this helix against the synaptobrevin and syntaxin-1 SNARE motifs. Cross-peaks from the accessory helix, which does not contact the SNARE complex, are also perturbed by binding because the stabilization of the central helix propagates toward the accessory helix, but the perturbations are progressively smaller toward the N-terminus due to fraying of the helix and retention of the intrinsic flexibility characteristic of the isolated CpxI(26-83) fragment (*Chen et al., 2002*).

To test whether the CpxI accessory helix can replace in part or in full the C-terminus of the synaptobrevin SNARE motif in the SNARE complex, we acquired $^1$H-$^{15}$N TROSY-HSQC spectra of $^2$H,$^{15}$N-labeled CpxI(26-83) bound to SNARE complexes where the synaptobrevin SNARE motif was truncated at residue 62 or 68 (SCΔ62 or SCΔ68). Comparison with the spectrum obtained in the presence of non-truncated SNARE complex (*Figure 2C*) showed that the well-resolved cross-peaks from several residues of the central helix (e.g., those of E58, E60, M62, R63 and Q64) moved gradually to the center of the spectrum, toward their positions in free CpxI(26-83), as the truncation was more severe. Moreover, some of these cross-peaks exhibited broadening that most likely arises from chemical exchange. In contrast, cross-peaks from more C-terminal residues of the central helix (e.g., those of I66, D68 and K69) were less affected by the truncations. This behavior is illustrated by the chemical

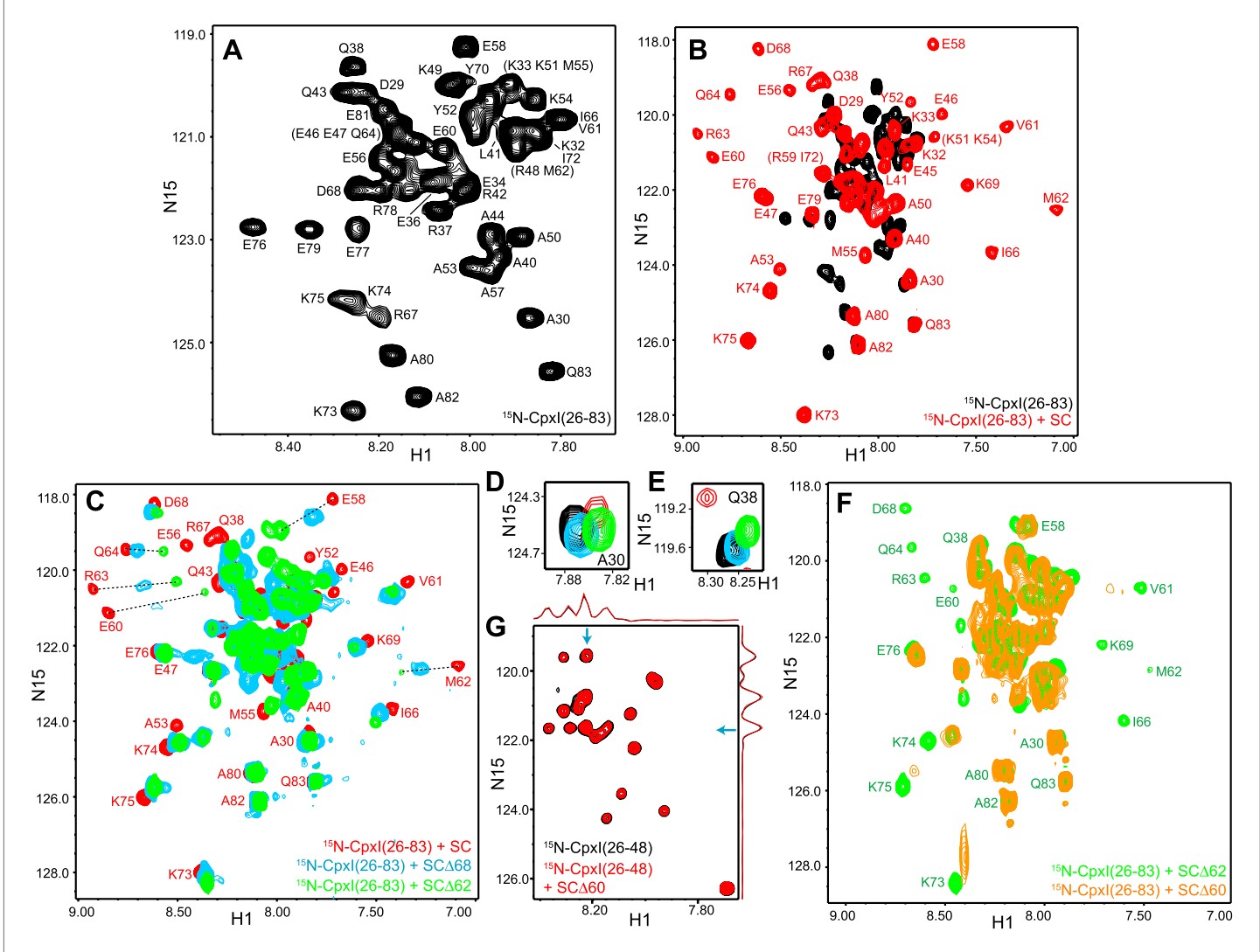

**Figure 2**. NMR analysis of interactions between $^2$H,$^{15}$N-labeled CpxI fragments and synaptobrevin-truncated SNARE complexes. (**A–C**) Expansions of $^1$H-$^{15}$N TROSY-HSQC spectra of $^2$H,$^{15}$N-CpxI(26-83) free (black contours), or bound to non-truncated SNARE complex (red contours), to SCΔ68 (blue contours) or SCΔ62 (green contours). Cross-peaks assignments for the free form are based on those described for full-length CpxI (*Pabst et al., 2000*) and assignments for CpxI(26-83) bound to non-truncated SNARE complex were described previously (*Chen et al., 2002*). In (**B** and **C**), the minimal contour levels of the different spectra were adjusted to enable visualization of the weakest cross-peaks of interest; hence, cross-peak intensities are not directly comparable. (**D** and **E**) Expansions of the regions containing the cross-peaks of A30 (**D**) or Q38 (**E**) of the spectra shown in panels (**A–C**). The minimal contour levels of the different spectra were adjusted to make the cross-peak intensities directly comparable. (**F**) Expansions of $^1$H-$^{15}$N TROSY-HSQC spectra of $^2$H,$^{15}$NCpxI(26-83) bound to SCΔ62 (green contours) or SCΔ60 (orange contours). (**G**) Expansions of $^1$H-$^{15}$N TROSY-HSQC spectra of WT $^2$H,$^{15}$N-CpxI(26-48) in the absence (black contours) or presence (red contours) of SCΔ60. Because the red and black spectra are practically identical, the black spectrum was plotted at slightly lower levels to facilitate observation of the black crosspeaks behind the red ones. However, the intensities of all the cross-peaks were the same in the black and red spectra within experimental error, as illustrated by the one-dimensional traces shown above and on the right of the two-dimensional contour plots (taken at the chemical shifts indicated by the blue arrows).

The following figure supplements are available for figure 2:

**Figure supplement 1**. Additional analysis of the interaction between $^2$H,$^{15}$N-labeled CpxI fragments and synaptobrevin-truncated SNARE complexes.

**Figure supplement 2**. NMR analysis of interactions between $^2$H,$^{15}$N-labeled CpxI superclamp mutant fragments and synaptobrevin-truncated SNARE complexes.

shift changes observed in the CpxI(26-83) central helix when comparing SC-bound vs SCΔ68-bound spectra [ΔδCpx(SCΔ68-SC)], normalized by the changes observed between free and SC-bound Cpx(26-83) [ΔδCpx(SC-free)] (*Figure 2—figure supplement 1A*). These ratios provide a measure of the destabilization of the central helix caused by the synaptobrevin C-terminal truncation. A plot of ΔδCpx(SCΔ68-SC) vs ΔδCpx(SC-free) (*Figure 2—figure supplement 1B*) also shows that cross-peaks from the C-terminus of the central helix were less affected by the C-terminal truncation.

The differential effects of the truncations show that the movements toward the center of the spectrum do not result simply from incomplete binding of the CpxI(26-83) fragment to the truncated SNARE complexes, which should be almost quantitatively bound based on affinities measured by ITC (see below). Instead, these data indicate that there is exchange between the normal bound state with a stable central helix and one or more states where the C-terminus of the central helix is stably packed against the SNAREs but the N-terminus of the central helix is flexible. These states become more populated for the Δ62 truncation than for Δ68. The effect of the Δ68 truncation can be attributed to an overall destabilization of the synaptobrevin helix beyond R56, the residue in the central polar layer of the SNARE complex that provides an approximate point of separation for two folding units corresponding to the N- and C-terminal halves of the complex (*Sorensen et al., 2006*; *Gao et al., 2012*). This destabilization is manifested in the $^1$H-$^{15}$N TROSY-HSQC spectra of the truncated SNARE complex described below and is transferred to the CpxI central helix, which is not surprising because the central helix makes extensive contacts with synaptobrevin residues spanning from R47 to A69 in the non-truncated complex (*Figure 1—figure supplement 1*). The stronger effects on the cross-peaks of the CpxI central helix caused by the Δ62 truncation, compared to Δ68, arise naturally from the removal of key synaptobrevin residues that interact with CpxI, including D64, D65 and D68.

Changes in the cross-peaks corresponding to the CpxI accessory helix caused by the Δ62 and Δ68 synaptobrevin truncations were more difficult to monitor because the cross-peaks are mostly located in the crowded center of the spectrum. The truncations did cause some changes in the center of the spectrum, but the number of cross-peaks and their overall distribution remained similar (*Figure 2C*). Cross-peaks from the accessory helix that could be identified in all the spectra exhibited some shifts in the different complexes, but in all cases they remained close to the position of the cross-peak corresponding to free CpxI(26-83) (illustrated by the A30 and Q38 cross-peaks in *Figure 2D,E*). Note that these shifts can be induced by changes in the stability of the helix in the different complexes and that insertion of the accessory helix into the truncated SNARE complexes is expected to induce much more dramatic shifts. Moreover, such insertion should cause strong broadening in the cross-peaks from the accessory helix, but the intensities of these cross-peaks actually increased in the spectrum of Cpx(26-83) bound to SCΔ68 with respect to the SC-bound state, and increased somewhat more in the SCΔ62-bound spectrum (*Figure 2D,E*, *Figure 2—figure supplements 1C,D*). These data show that the synaptobrevin C-terminal truncations increase the flexibility of the accessory helix, in correlation with the destabilization of N-terminal part of the central helix, and provide very strong evidence against the notion that the accessory helix of CpxI(26-83) inserts into the grove generated by the truncations.

Since the crystal structure of the CpxI(26-83) D27L, E34F, R37A superclamp mutant bound to a SNARE complex with C-terminally synaptobrevin (*Figure 1B*) was obtained with a complex containing synaptobrevin truncated at residue 60 (SCΔ60), we also acquired $^1$H-$^{15}$N TROSY-HSQC spectra of $^2$H,$^{15}$N-CpxI(26-83) in the presence of SCΔ60. The spectra was similar to that obtained with SCΔ62, but most cross-peaks from the central helix were broadened beyond detection (*Figure 2F*). This behavior can be attributed to stronger chemical exchange broadening, which is particularly well manifested for the cross-peaks of K73 and K75 (which are adjacent to the central helix). The well-resolved cross-peaks from the accessory helix of CpxI(26-83) bound to SCΔ60 (e.g., those of A30 and Q38) had similar intensities as those observed upon binding to SCΔ62, showing that the accessory helix does not insert into the groove left by the Δ60 truncation. Because the zigzag array observed in the crystal structure of the CpxI(26-83) superclamp/SCΔ60 complex suggested that the accessory helix should be able to bind by itself to SCΔ60, without the central helix, we also acquired $^1$H-$^{15}$N TROSY-HSQC spectra of a WT $^2$H,$^{15}$N-CpxI fragment spanning the accessory helix [CpxI(26-48)]. SCΔ60 did not cause substantial changes in the spectra of CpxI(26-48) (*Figure 2G*). Because of the very high sensitivity of these spectra to binding to protein complexes such as SCΔ60, particularly for flexible peptides such as CpxI(26-48), even a small percentage of binding should be reflected in some cross-peak broadening. Hence, these results clearly show that CpxI(26-48) does not bind to the synaptobrevin Δ60 truncated SNARE complex in solution under the conditions of our experiments.

We also acquired parallel $^1$H-$^{15}$N TROSY-HSQC spectra of $^2$H,$^{15}$N-labeled fragments of the CpxI(26-83) D27L, E34F, R37A superclamp mutant (supcl) in the presence and absence of SCΔ60. The data acquired with $^2$H,$^{15}$N-CpxI(26-48)supcl showed no binding to SCΔ60 (*Figure 2—figure supplement 2A*), as observed for WT CpxI(26-48). The spectrum obtained for $^2$H,$^{15}$N-CpxI(26-83)supcl bound to SCΔ60 was similar to that obtained for WT CpxI(26-83), with a few differences that arise from the mutations (*Figure 2—figure supplement 2B*) and are also observed in the spectra obtained for the free CpxI fragments (*Figure 2—figure supplement 2C*). Moreover, the overall effects of SCΔ60 binding to CpxI(26-83)suplc are similar to those observed for WT CpxI(26-83) (*Figure 2—figure supplement 2B,D,E*) and, as observed for the WT protein, the cross-peaks from the accessory helix do not exhibit dramatic shifts and/or broadening as would be expected for insertion into the groove left by the synaptobrevin truncation. Therefore, we were unable to detect an interaction between the accessory helix of CpxI(26-83)supcl and SCΔ60 under the conditions of our NMR experiments, although we cannot rule out the possibility that there is a weak interaction in solution that becomes stabilized by crystallization.

## The accessory helix does not insert into synaptobrevin-truncated SNARE complexes: NMR analysis with $^2$H,$^{15}$N-labeled SNARE complexes

To further test the insertion model, we also acquired $^1$H-$^{15}$N TROSY-HSQC spectra of truncated SNARE complexes that were $^2$H,$^{15}$N-labeled at the C-terminal SNARE motif of SNAP-25 (SNC) or at the syntaxin-1 SNARE motif (Syx), since these SNARE motifs were predicted to contact the CpxI accessory helix in synaptobrevin-truncated SNARE complexes. We first compared spectra of $^2$H,$^{15}$N-SNC complexes that were non-truncated or truncated at residues 62 or 76 of synaptobrevin ($^2$H,$^{15}$N-SNC-SCΔ62 or $^2$H,$^{15}$N-SNC-SCΔ76) (*Figure 3A,B*), and found that progressive truncation led to increased appearance of sharp cross-peaks in the center of the spectrum and disappearance of cross-peaks from the SNC C-terminal residues in well-resolved regions, or shifts for residues close to the polar layer (Q174 for SNC). These results show that the synaptobrevin truncations lead to flexibility in the C-terminal half of SNC. For the Δ62 truncation, stable structure appears to remain only up to residue 180.

CpxI(26-83) caused only small shifts in some of the well-resolved cross-peaks of the $^1$H-$^{15}$N TROSY-HSQC spectrum of $^2$H,$^{15}$N-SNC-SCΔ62 (*Figure 3C*), in correlation with the small perturbations observed for the non-truncated SNARE complex because CpxI makes little contact with SNAP-25 (*Chen et al., 2002*). Moreover, these small shifts can arise from stabilization of the synaptobrevin and syntaxin-1 helix in the truncated SNARE complex upon binding to CpxI(26-83). Importantly, CpxI(26-83) binding induced practically no changes in the cross-peaks corresponding to the flexible C-terminus of SNC in the Δ62 SNARE complex (*Figure 3C*), showing again that the CpxI accessory helix does not interact with the truncated SNARE complex. We also analyzed perturbations caused by the Cpx(26-83)supcl mutant or a fragment spanning the accessory and central helices of CpxI from *Drosophila Melanogaster* [dmCpxI(28-88)], which inhibits release more strongly than mammalian CpxI (*Huntwork and Littleton, 2007*; *Xue et al., 2009*), but the results (*Figure 3—figure supplement 1A,B*) were analogous to those obtained with WT CpxI(26-83). Since the crystal structure leading to the zig-zag model (*Figure 1B*) was obtained with synaptobrevin truncated at residue 60, we acquired additional $^1$H-$^{15}$N TROSY-HSQC spectra with $^2$H,$^{15}$N-SNCΔ60, but similar data were obtained again on addition of WT CpxI(26-83) or Cpx(26-83)supcl mutant (*Figure 3—figure supplement 1C,D*). All these results suggest that the accessory helix of WT CpxI(26-83), Cpx(26-83)supcl and dmCpx(28-88) do not insert into the SNARE complexes containing C-terminally truncated synaptobrevin. Finally, addition of a CpxI fragment spanning only the accessory helix [CpxI(26-48)] caused practically no changes on the $^1$H-$^{15}$N HSQC spectrum of $^2$H,$^{15}$N-SNC-SCΔ62 (*Figure 3D*), confirming that the accessory helix does not bind by itself to the truncated SNARE complex or binds with extremely weak affinity.

We also performed parallel experiments with truncated SNARE complex $^2$H,$^{15}$N-labeled at the syntaxin-1 SNARE motif ($^2$H,$^{15}$N-Syx-SCΔ62). Comparison of the $^1$H-$^{15}$N TROSY HSQC spectra of this complex and the non-truncated complex (*Figure 3E*) revealed that the Δ62 truncation led to disappearance of most of the well-resolved cross-peaks from residues beyond the residue in the polar layer (Q226 for Syx) and an increase in sharp cross-peaks in the center of the spectrum. These results correlated with those obtained with $^2$H,$^{15}$N-SNC-SCΔ62 and indicate that the C-terminal half of the syntaxin-1 SNARE motif becomes flexible due to the Δ62 synaptobrevin truncation, although the smaller number of cross-peaks in the middle suggests that there may be broadening due to exchange between structured and flexible conformations. CpxI(26-83) caused multiple changes in the $^1$H-$^{15}$N TROSY HSQC

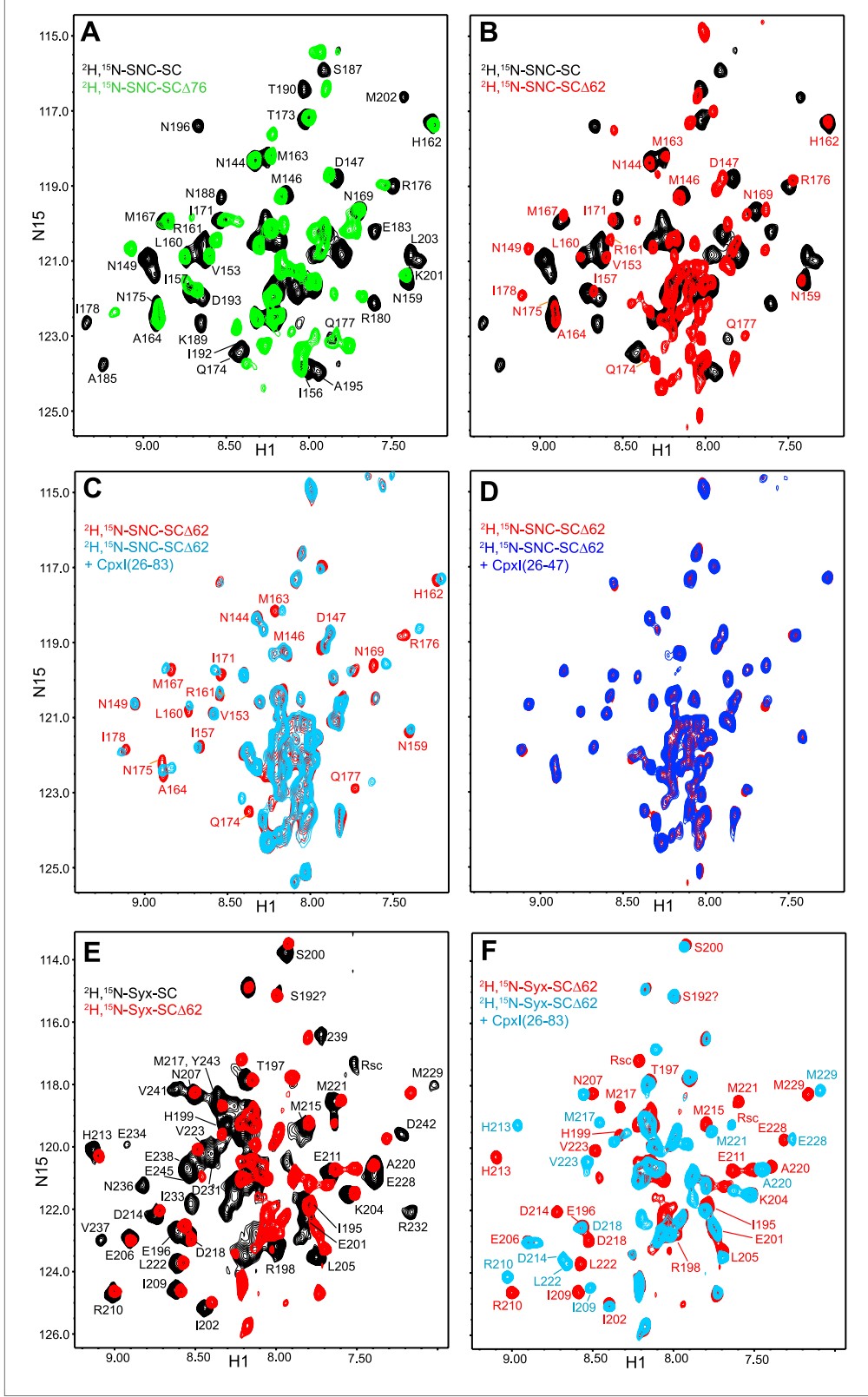

**Figure 3**. NMR analysis of interactions between ²H,¹⁵N-labeled synaptobrevin-truncated SNARE complexes and CpxI fragments. (**A** and **B**) Expansions of ¹H-¹⁵N TROSY-HSQC spectra of the non-truncated ²H,¹⁵N-SNC-SC (black contours), ²H,¹⁵N-SNC-SCΔ76 (green contours) and ²H,¹⁵N-SNC-SCΔ62 (red contours). (**C** and **D**) Expansions of ¹H-¹⁵N

*Figure 3. Continued on next page*

*Figure 3. Continued*

TROSY-HSQC spectra of $^2$H,$^{15}$N-SNC-SCΔ62 alone (red contours) or in the presence of CpxI(26-83) (light blue; **C**) or CpxI(26-47) (dark blue; **D**). (**E** and **F**) Expansions of $^1$H-$^{15}$N TROSY-HSQC spectra of $^2$H,$^{15}$N-Syx-SC (black contours) and of $^2$H,$^{15}$N-Syx-SCΔ62 in the absence (red contours) or presence (light blue contours) of CpxI(26-83). Cross-peak assignments are based on those described for the non-truncated SNARE complex (***Chen et al., 2002***; ***Chen et al., 2005***).

The following figure supplements are available for figure 3:

**Figure supplement 1**. Additional NMR analyses of interactions between $^2$H,$^{15}$N-labeled synaptobrevin-truncated SNARE complexes and CpxI fragments.

spectrum of the $^2$H,$^{15}$N-Syx-SCΔ62 complex (***Figure 3F***) that correlate with those observed for the non-truncated SNARE complex (***Chen et al., 2002***). Only limited changes were observed for the sharp cross-peaks in the middle of the spectrum, which can arise from partial stabilization of the syntaxin-1 helix. Although these data are less conclusive than those obtained with the $^2$H,$^{15}$N-SNC-SCΔ62 complex, it is clear that CpxI(26-83) binding did not yield new well-dispersed cross-peaks that might correspond to structured syntaxin-1 C-terminal residues interacting with the CpxI(26-83) accessory helix. Hence, these data further support the conclusion that the accessory helix does not interact with C-terminally synaptobrevin truncated SNARE complex.

## The accessory helix does not insert into synaptobrevin-truncated SNARE complexes: analysis by ITC

The primary evidence reported to support the notion that the accessory helix of WT CpxI(26-83) inserts into the synaptobrevin Δ60 truncated SNARE complex, as observed by crystallography for the CpxI(26-83) D27L, E34F, R37A superclamp mutant, was obtained in competition assays monitored by ITC (***Kummel et al., 2011***). In these assays, 1.5 equivalents of CpxI lacking the accessory helix [CpxI(47-134)] were used to block the central helix binding site of SCΔ60, and the heat observed on addition of CpxI(26-83) was attributed to binding of the accessory helix of CpxI(26-83) to SCΔ60. This interpretation assumed that 1.5 equivalents for CpxI(47-134) were sufficient to quantitatively saturate SCΔ60, but the validity of this assumption is unclear because removal of multiple synaptobrevin residues that interact with CpxI in the non-truncated SNARE complex is expected to considerably decrease the affinity of SCΔ60 for CpxI.

To address this issue, we measured the affinity of CpxI(47-134) for SCΔ60 and the non-truncated SNARE by ITC. For the latter (***Figure 4A***), we measured a $K_d$ of 339 ± 9 nM ($\Delta H$ = −32.6 kcal/mol; N = 0.95), which is higher than that we obtained for CpxI(26-83) [$K_d$ = 25.3 nM; ***Xu et al., 2013***] and may arise because of favorable long-range electrostatic interactions between the accessory helix and the SNARE complex. Importantly, binding of Cpx(47-134) to SCΔ60 was even weaker, with a $K_d$ of 2.39 ± 0.19 μM ($\Delta H$ = −19.5 kcal/mol; N = 0.92) (***Figure 4B***). This decreased affinity implies that binding of CpxI(47-134) to SCΔ60 is not saturated upon addition of 1.5 equivalents of CpxI(47-134) (arrow in ***Figure 4B***). Hence, the heat observed upon addition of CpxI(26-83) to SCΔ60 blocked with CpxI(27-134) arises from completion of the titration of the central helix binding site, rather than from interactions involving the accessory helix. We confirmed this conclusion by titrating CpxI(47-134) itself on a sample containing SCΔ60 and 1.5 equivalents of CpxI(47-134) (***Figure 4C***), which logically yielded data similar to those observed in the direct titration experiment of ***Figure 4B*** beyond 1.5 equivalents. Moreover, adding CpxI(26-83) to SCΔ60 prebound to 1.5 equivalents of CpxI(47-134) yielded very similar results (***Figure 4D***), which in turn were also comparable to the data described in ***Kummel et al. (2011)***. To test for binding of the accessory helix under conditions where the central helix binding site was more saturated, we performed experiments with SCΔ60 prebound to 3.0 equivalents of CpxI(47-134) (93% binding based on the $K_d$ described above). As expected, addition of Cpx(26-83) yielded only a small amount of heat that again is the natural extension of the direct titration of the central helix binding site (***Figure 4—figure supplement 1***). Hence, no binding of the accessory helix of Cpx(26-83) to SCΔ60 is detected in these experiments.

The equations describing competition data are more complicated than those describing a single-site binding model, but we used this simplified model to fit our ITC data to allow comparison with the results described in ***Kummel et al. (2011)***. We obtained apparent $K_d$ = 3.73 ± 0.57 μM and apparent $\Delta H$ = −5.5 kcal/mol for competition with CpxI(47-134) (***Figure 4C***), and apparent $K_d$ = 5.41 ± 1.48 μM

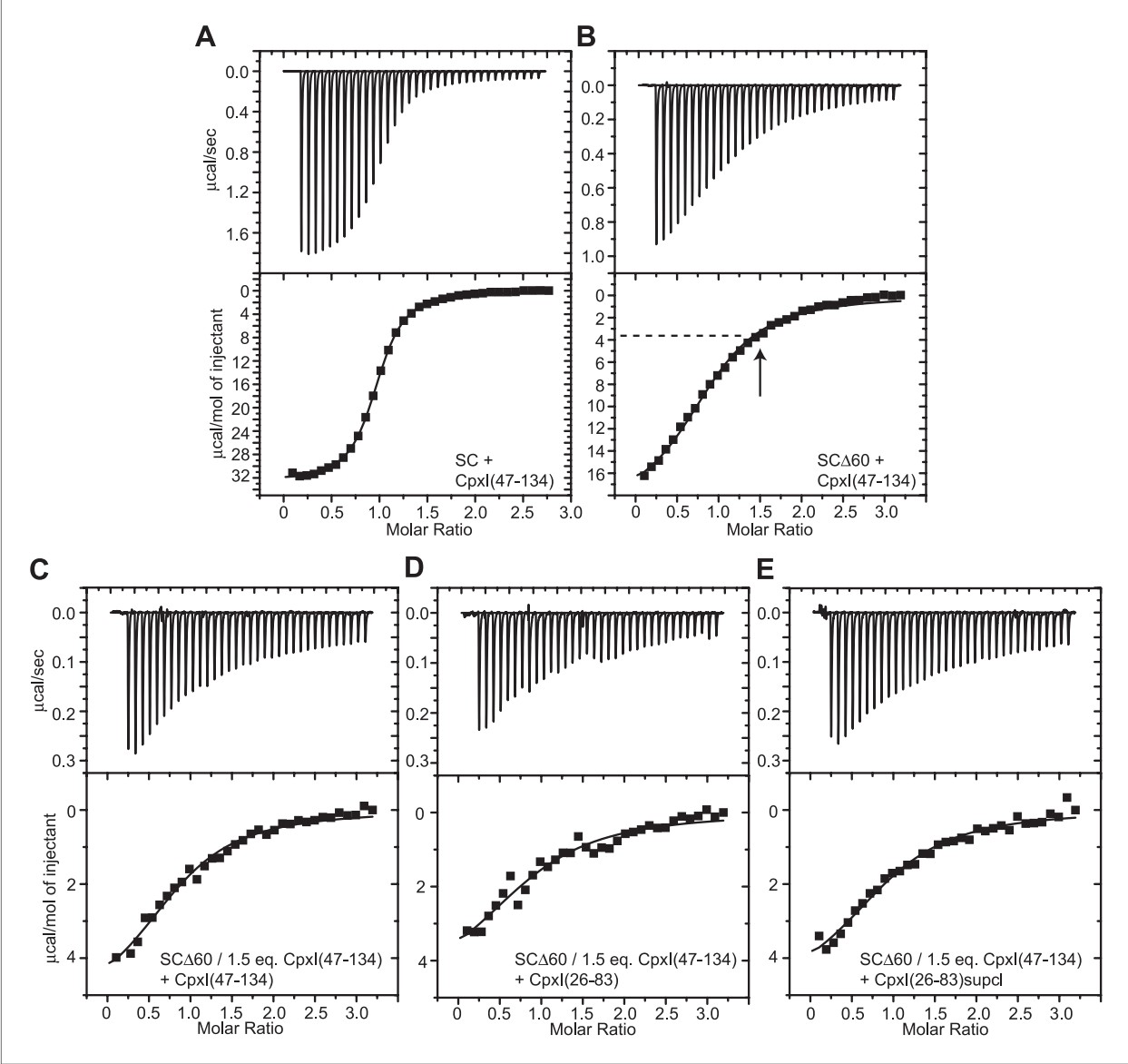

**Figure 4**. ITC analysis of binding of CpxI fragments to SNARE complexes. (**A** and **B**) Direct titrations of non-truncated SNARE complex (SC; **A**) or SCΔ60 (**B**) with CpxI(47-134). (**C–E**) Competition assays where samples containing SCΔ60 and 1.5 equivalents of CpxI(47-134) were titrated with CpxI(47-134) (**C**), WT CpxI(26-83) (**D**) or CpxI(26-83) D27L, E34F, R37A superclamp mutant (supcl) (**E**). The arrow in panel (**B**) shows the point of the direct titration where 1.5 equivalents of Cpx(47-134) had been added, and the dashed line shows the heat measured at that point of the titration. The same heat (within experimental error) was measured at the start of the competition experiments of panels (**C–E**). Thus, the heat measurements in the competition assays correspond to the completion of the titration (i.e., the tail of the direct titration of panel **B**) because 1.5 equivalents of CpxI(47-134) were not sufficient to saturate SCΔ60.

The following figure supplements are available for figure 4:

**Figure supplement 1**. Additional ITC analysis of binding of CpxI fragments to SNARE complexes.

and apparent ΔH = −5.0 kcal/mol for competition with CpxI(26-83) (***Figure 4D***), which are clearly similar. ***Kummel et al. (2011)*** reported $K_d$ = 16 μM and ΔH = −5.4 kcal/mol for competition with CpxI(26-83). Although the $K_d$ is somewhat different, we consider that our data do reproduce the results of ***Kummel et al. (2011)*** within experimental error, considering the approximation involved in the single-site model. However, our results show that the interpretation of the competition ITC assays needs to be revised and that these assays are unable to detect an interaction between the accessory helix of WT CpxI(26-83) and SCΔ60, in agreement with our NMR results.

We also performed competition assays with CpxI(26-83) D27L, E34F, R37A superclamp mutant and obtained very similar results to those observed with WT CpxI(26-83) (*Figure 4E*; apparent $K_d$ = 4.03 ± 0.89 µM; apparent ΔH = −5.1 kcal/mol). These results contrast with those described previously (*Kummel et al., 2011*) but agree with our NMR data and further suggest that even the accessory helix of the CpxI(26-83)supcl mutant does not interact with SNARE complexes containing C-terminally truncated synaptobrevin in solution or, if there is any interaction, it is weak and cannot be detected in our NMR and ITC experiments.

## Analysis of interactions between CpxI and syntaxin-1-truncated SNARE complexes

We also investigated whether the complexin accessory helix might inhibit neurotransmitter release by replacing the syntaxin-1 SNARE motif in partially assembled SNARE complexes. For this purpose, we first acquired $^1$H-$^{15}$N TROSY-HSQC spectra of $^2$H,$^{15}$N-labeled synaptobrevin SNARE motif ($^2$H,$^{15}$N-Syb) free and incorporated into non-truncated SNARE complex ($^2$H,$^{15}$N-Syb-SC) or SNARE complex with syntaxin-1 truncated at residue 236 ($^2$H,$^{15}$N-Syb-SCΔ236). Comparison of the spectra obtained for free $^2$H,$^{15}$N-Syb and $^2$H,$^{15}$N-Syb-SC showed again the dramatic spectral changes that occur when a flexible sequence such as $^2$H,$^{15}$N-Syb forms a stable complex (*Figure 5A*; cross-peak assignments for $^2$H,$^{15}$N-Syb based on those of Syb[1-96] [*Hazzard et al., 1999*] are shown in *Figure 5—figure supplement 1A*). The truncation of the syntaxin-1 C-terminus in $^2$H,$^{15}$N-Syb-SCΔ236 led to disappearance of most well-dispersed cross-peaks from the C-terminal half of the synaptobrevin SNARE motif and appearance of new, sharp cross-peaks in the middle of the spectrum (*Figure 5B*). We obtained assignments for some of these cross-peaks using triple resonance experiments as described (*Chen et al., 2002*) and found that they generally were at similar positions to those observed for free $^2$H,$^{15}$N-Syb (e.g., for A67, A69, A72, A74, W89 and K91; compare *Figure 5B*, orange contours, with *Figure 5—figure supplement 1A*). These results suggest that the C-terminal half of the synaptobrevin SNARE motif is flexible in $^2$H,$^{15}$N-Syb-SCΔ236 as a result of the syntaxin-1 truncation. We also acquired $^1$H-$^{15}$N TROSY-HSQC spectra for the same truncated complex but with the SNAP-25 C-terminal motif $^2$H,$^{15}$N-labeled ($^2$H,$^{15}$N-SNC-SCΔ236) and again found disappearance of the well-dispersed cross-peaks from the C-terminal half of SNC with concomitant appearance of sharp cross-peaks in the middle of the spectrum (*Figure 5—figure supplement 1B*). These changes are similar to those caused by the synaptobrevin truncation in $^2$H,$^{15}$N-SNC-SCΔ62 (*Figure 3B*) and show that the C-terminus of SNC also becomes flexible upon truncation of syntaxin-1.

We next examined the changes induced by CpxI(26-83) on the $^1$H-$^{15}$N TROSY-HSQC spectra of these complexes. For non-truncated complex ($^2$H,$^{15}$N-Syb-SC), CpxI(26-83) caused multiple cross-peak shifts, particularly for synaptobrevin residues that contact the CpxI central helix (*Figure 5C*), as observed previously (*Chen et al., 2002*). Binding to CpxI(26-83) induced similar shifts for those residues in $^2$H,$^{15}$N-Syb-SCΔ236, but in addition caused disappearance of many sharp cross-peaks in the middle of the spectrum that correspond to the flexible synaptobrevin C-terminal half (*Figure 5D*). As a result, the $^1$H-$^{15}$N TROSY HSQC spectrum of $^2$H,$^{15}$N-Syb-SCΔ236 bound to CpxI(26-83) is similar to that obtained for the non-truncated SNARE complex except that most cross-peaks corresponding to the synaptobrevin C-terminal half disappeared (*Figure 5E*). Such disappearance most likely arises from chemical exchange between the flexible conformations characteristic of the synaptobrevin C-terminal half in $^2$H,$^{15}$N-Syb-SCΔ236 and a more defined structure(s) induced upon CpxI(26-83) binding. These results contrast with those obtained upon addition of CpxI(26-83) to $^2$H,$^{15}$N-SNC-SCΔ236 (*Figure 5—figure supplement 1C*), which revealed only small cross-peak shifts for a few well-dispersed cross-peaks and no marked changes for the sharp cross-peaks in the middle of the spectrum, as observed for the synaptobrevin-truncated SNARE complex $^2$H,$^{15}$N-SNC-SCΔ62 (*Figure 3C*). We also analyzed the effects of CpxI(26-47) on the $^1$H-$^{15}$N TROSY-HSQC spectrum of $^2$H,$^{15}$N-SNC-SCΔ236 but observed practically no changes (*Figure 5F*), showing that the accessory helix by itself does not bind to the syntaxin-1 truncated SNARE complex.

While the results obtained with $^2$H,$^{15}$N-SNC-SCΔ236 show that the accessory helix of CpxI(26-83) does not interact with the SNAP-25 C-terminus in the syntaxin-1 truncated SNARE complex and hence does not insert into the complex, the data acquired with $^2$H,$^{15}$N-Syb-SCΔ236 suggested that the accessory helix might interact with the synaptobrevin C-terminal half, which could provide an alternative mechanism to hinder SNARE complex assembly and thus inhibit neurotransmitter release (*Figure 1D*). To test this possibility, we analyzed the effects of SCΔ236 and other SNARE complexes with stronger

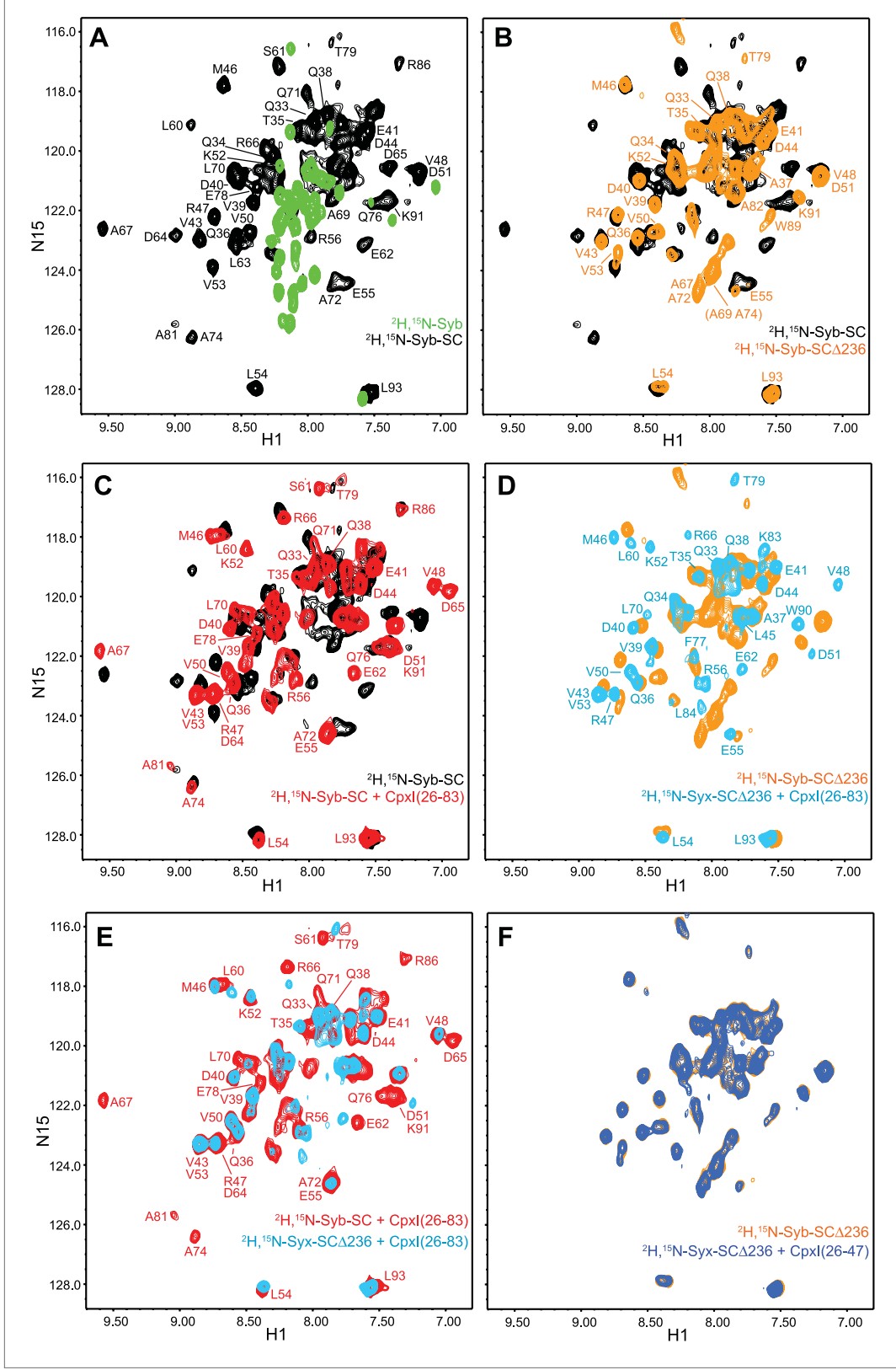

**Figure 5**. NMR analysis of interactions between ²H,¹⁵N-labeled syntaxin-1 truncated SNARE complexes and CpxI fragments. (**A** and **B**) Expansions of ¹H-¹⁵N TROSY-HSQC spectra of free ²H,¹⁵N-labeled synaptobrevin SNARE motif (²H,¹⁵N-Syb, green contours), the non-truncated ²H,¹⁵N-Syb-SC (black contours) and ²H,¹⁵N-Syb-SCΔ236 (orange contours).
*Figure 5. Continued on next page*

*Figure 5. Continued*

(**C**) Expansions of $^1$H-$^{15}$N TROSY-HSQC spectra of $^2$H,$^{15}$N-Syb-SC in the absence (black contours) and presence (red contours) of CpxI(26-83). (**D**) Expansions of $^1$H-$^{15}$N TROSY-HSQC spectra of $^2$H,$^{15}$N-Syb-SCΔ236 in the absence (orange contours) and presence (light blue contours) of CpxI(26-83). (**E**) Superposition of expansions of $^1$H-$^{15}$N TROSY HSQC spectra of $^2$H,$^{15}$N-Syb-SC (red contours) and $^2$H,$^{15}$N-Syb-SCΔ236 (light blue contours) bound to CpxI(26-83). (**F**) Expansions of $^1$H-$^{15}$N TROSY-HSQC spectra of $^2$H,$^{15}$N-Syb-SCΔ236 in the absence (orange contours) and presence (dark blue contours) of CpxI(26-47). Cross-peaks assignments for $^2$H,$^{15}$N-Syb-SC free and bound to CpxI(26-83) were described previously (***Chen et al., 2002***). Cross-peaks assignments for $^2$H,$^{15}$N-Syb-SCΔ236 that were not immediately clear from those obtained for $^2$H,$^{15}$N-Syb-SC were obtained using triple resonance experiments.

The following figure supplements are available for figure 5:

**Figure supplement 1**. Additional NMR analyses of interactions between $^2$H,$^{15}$N-labeled syntaxin-1 truncated SNARE complexes and CpxI fragments.

---

truncations in the syntaxin-1 C-terminus (SCΔ232 and SCΔ228) on the $^1$H-$^{15}$N TROSY-HSQC spectrum of $^2$H,$^{15}$N-labeled CpxI(26-83). The spectra obtained upon binding to SC or to SCΔ236 revealed only small shifts in a few well-resolved cross-peaks and practically no perturbations of the sharp cross-peaks in the center of the spectrum corresponding to the accessory helix (***Figure 6A***). When we included the spectra obtained in the presence of SCΔ232 and SCΔ228 in the comparison, it became clear that a few well-resolved cross-peaks (e.g., those of E60, M62, R63, and Q64) shift gradually to the center of the spectrum, toward their positions in free $^2$H,$^{15}$N-CpxI(26-83), as the truncation in syntaxin-1 is more severe (***Figure 6B***). This result is similar to that caused by truncations in the synaptobrevin C-terminus (***Figure 2C***) and can thus be attributed to increasing destabilization of the CpxI central helix as more residues are deleted in syntaxin-1 (the effects are smaller because most of the residues deleted in syntaxin-1 do not contact CpxI[26-83]; ***Figure 1—figure supplement 1***).

As in the case of the synaptobrevin truncations (***Figure 2C***), the syntaxin-1 truncations in the SNARE complex also caused changes in the center of the $^1$H-$^{15}$N TROSY-HSQC spectrum of the bound $^2$H,$^{15}$N-CpxI(26-83), but the number of cross-peaks and their overall distribution remained similar (***Figure 6B***) and the well-resolved cross-peaks corresponding to A30 and Q38 exhibited small gradual shifts from their positions upon binding to the non-truncated SNARE complex to their free positions as syntaxin-1 was increasingly truncated (***Figure 6C,D***). These results strongly suggest that the shifts in the CpxI(26-83) accessory helix do not arise from interactions with the C-terminal half of the synaptobrevin SNARE motif but rather because the destabilization of the central helix caused by the truncations in syntaxin-1 is transferred into destabilization of the accessory helix. Note also that no substantial broadening of the cross-peaks from the accessory helix was observed, in contrast to the broadening beyond disappearance of cross-peaks from the synaptobrevin C-terminal half in $^2$H,$^{15}$N-Syb-SCΔ236 bound to CpxI(26-83) (***Figure 5D,E***). It is likely that such disappearance arises from chemical exchange between the flexible conformations of the synaptobrevin C-terminal half in free $^2$H,$^{15}$N-Syb-SCΔ236 and formation of a defined helical structure that does not contact CpxI(26-83) but is nucleated by stabilization of the middle of synaptobrevin upon binding of the CpxI(26-83) central helix. Regardless of the validity of this explanation, it is clear from the behavior of the cross-peaks of the accessory helix that this helix does not interact with the synaptobrevin SNARE motif of the syntaxin-1 truncated SNARE complexes, arguing against the model of ***Figure 1D***.

## Complexin I superclamp mutations do not inhibit neurotransmitter release

In parallel with our structural studies, we tested the complexin insertion and zigzag models emerging from cell–cell fusion assays (***Giraudo et al., 2006***) and X-ray crystallography (***Kummel et al., 2011***) by examining the effects on neurotransmitter release of some of the mutations that were reported to alter the inhibitory activity of CpxI in cell–cell fusion. In particular, we analyzed the effects of the D27L, E34F, R37A superclamp mutation that strongly inhibited cell–cell fusion, and of a 'poor-clamp' mutation that decreased the clamping efficiency of CpxI (K26A) (***Figure 7A***). Note that rescue studies on complexin knockdown cortical neurons found that the CpxI D27L, E34F, R37A superclamp mutation did not alter evoked neurotransmitter release and appeared to induce a modest decrease in spontaneous release, but it was unclear whether the decrease was significant, in part because of the small nature of the effect and in part because there was no direct comparison with a rescue using WT CpxI (***Yang et al., 2010***).

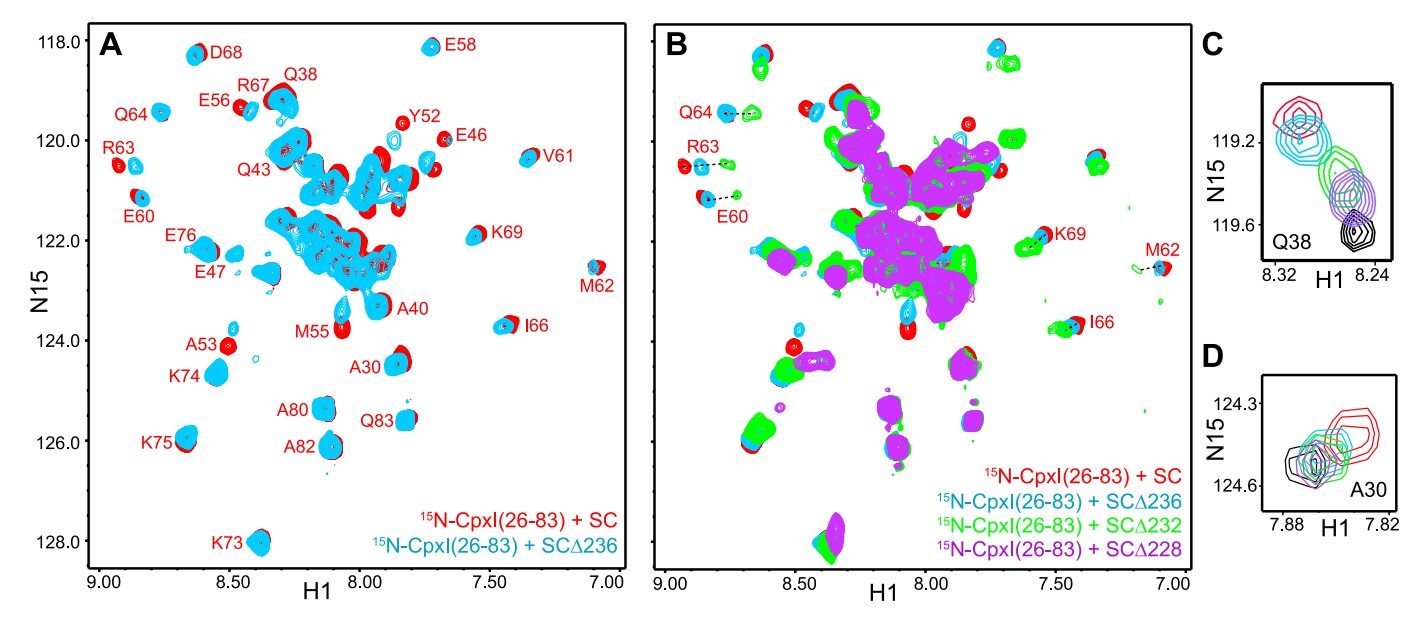

**Figure 6**. NMR analysis of interactions between $^2$H,$^{15}$N-labeled CpxI fragments and syntaxin-1-truncated SNARE complexes. (**A** and **B**) Expansions of $^1$H-$^{15}$N TROSY-HSQC spectra of $^2$H,$^{15}$N-CpxI(26-83) bound to nontruncated SNARE complex (SC; red contours), to SCΔ236 (blue contours), SCΔ232 (green contours) or SCΔ228 (purple contours). Cross-peaks assignments for CpxI(26-83) bound to nontruncated SNARE complex were described previously (**Chen et al., 2002**). (**C** and **D**) Expansions of the regions containing the cross-peaks of Q38 (**C**) or A30 (**D**) of the spectra shown in panels (**A** and **B**).

In our studies we performed rescue experiments on hippocampal glutamatergic neurons from complexin I-III triple KO mice. Expression was mediated via lentiviral transduction, and WT and mutant CpxI levels were monitored by immunocytochemistry and western blot analysis (**Figure 7—figure supplement 1**). CpxI-III-deficient neurons exhibit reduced vesicle released probability (Pvr), increased paired-pulse ratio (PPR), a facilitatory phenotype at high frequency stimulation and reduced spontaneous release frequency, all of which can be rescued with lenti-viral overexpression of WT CpxI (**Xue et al., 2008b**, **2009**). Hence, these synaptic parameters served us as readouts for the rescue behavior of different mutant Cplx variants.

The D27L, E34F, R37A superclamp mutation that introduces hydrophobic residues in the CpxI accessory helix did not change the ability to rescue the complexin I-III KO phenotype. On the contrary the amplitudes of action potential-evoked excitatory postsynaptic currents (EPSCs) tended to be larger that those observed upon rescue with WT CpxI (**Figure 7B**). As the size of the readily releasable pool (RRP) measured by hypertonic solution (**Rosenmund and Stevens, 1996**) was unchanged, the calculated Pvr was also slightly increased in the rescue with the superclamp CpxI mutants (**Figure 7C,D**). The facilitatory synaptic short-term plasticity behavior in complexin I-III triple KO neurons, measured through five EPSCs at 50 Hz and also reflected by calculation of the PPR (EPSC2/EPSC1), could be reversed to a more depressing phenotype in CpxI D27L, E34F, R37A expressing KO neurons compared to the rescue with WT CpxI (**Figure 7E**). Similarly, depression of EPSC amplitudes during a train of 50 evoked action potentials (AP) at 10 Hz was slightly stronger, and the spontaneous release of vesicles tended to be higher for the super-clamp mutant than for the WT rescue neurons (**Figure 7F,G**). Conversely, the 'non-clamping' K26A mutation impaired full rescue of the KO phenotype, as the EPSC amplitude and Pvr were reduced, and the PPR was increased (**Figure 7B–E**). The short-term plasticity experiment applying 50 AP at 10 Hz also revealed weak rescue activity (**Figure 7E,F**), but spontaneous release was not different from the WT rescue (**Figure 7G**).

Overall, these results are in contrast with key predictions from the insertion and zigzag models and do not correlate with the data obtained with cell–cell fusion assays (**Giraudo et al., 2009**), revealing an impairment of evoked release by the K26A mutation that decreases clamping activity in the cell–cell fusion assay and small effects with a tendency to increase evoked release for the superclamp mutant.

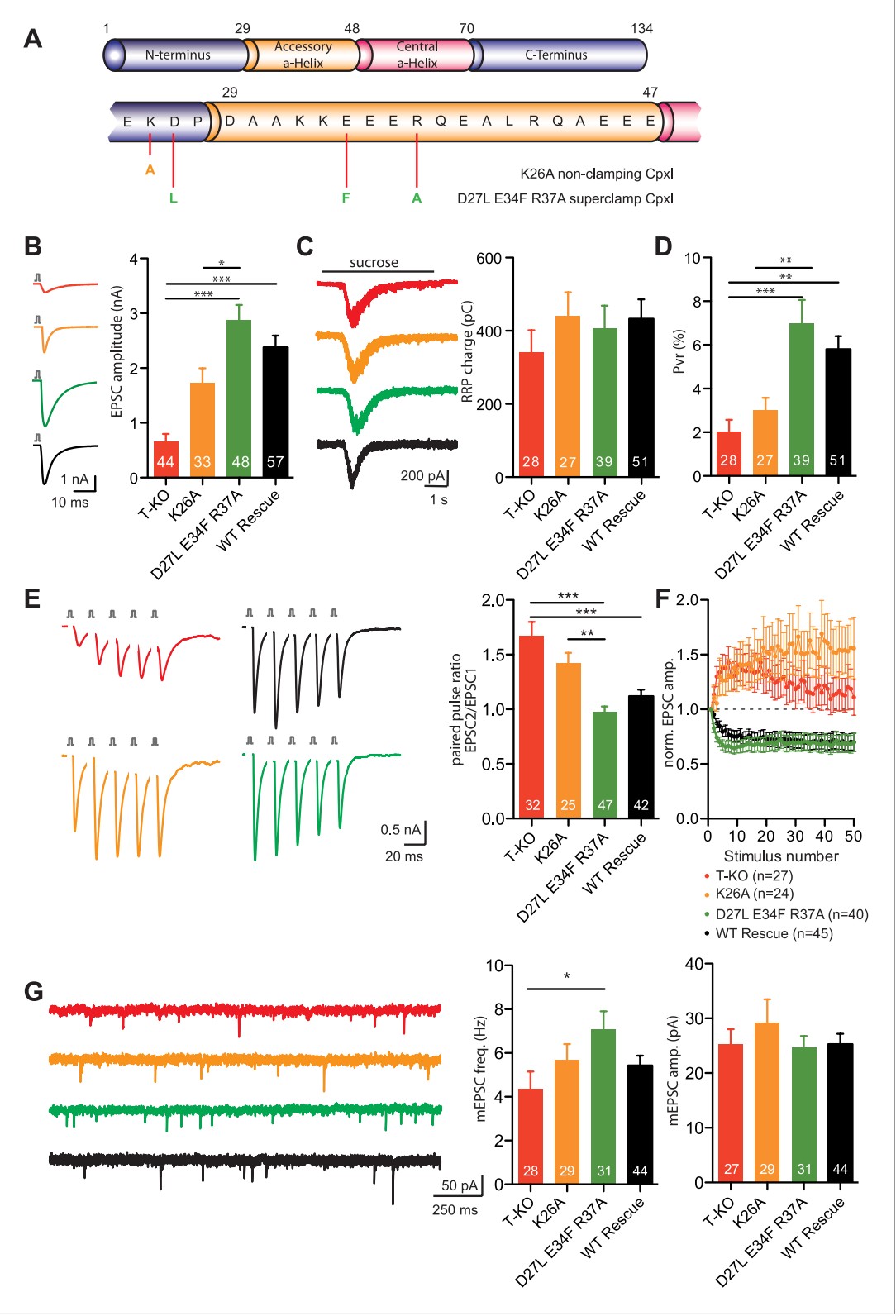

**Figure 7**. Rescue of the complexin KO phenotype with CpxI-superclamp, but not with a clamping deficient CpxI. (**A**) Overview of the introduced mutations in CpxI. (**B–C**) Representative traces and summary data of evoked EPSC (**B**) and synaptic responses to hypertonic sucrose solution (RRP) (**C**) of T-KO, K26A, D27L E34F R37A and WT-CplxI expressing hippocampal neurons. (**D**) Bar graph of the calculated vesicular release probability Pvr. (**E–F**) Analysis of

*Figure 7. Continued on next page*

*Figure 7. Continued*

short-term plasticity behavior: Example traces of a train of 5 APs at 50 Hz of T-KO, K26A, D27L E34F R37A and WT-CpxI expressing neurons from which the paired pulse ratio was calculated (**E**) and amplitudes of 50 EPSCs evoked at 10 Hz which were normalized to the first EPSCs and plotted over stimulus number (**F**). (**G**) Spontaneous transmitter release: Representative traces of T-KO, K26A, D27L E34F R37A and WT-CpxI expressing neurons and summary data of mEPSC frequency and mEPSC amplitude. Data are expressed as mean ± SEM, *p<0.05; **p<0.01; ***p<0.001. The numbers of neurons analyzed are shown within the bars. Vertical bars in the traces (**B** and **E**) represent 2-ms somatic depolarizations; depolarization artifact and action potentials were blanked. Time of sucrose application is indicated as horizontal line (**C**).

The following figure supplements are available for figure 7:

**Figure supplement 1**. Expression of Cpx variants in hippocampal CPXI-III triple KO neurons by lentiviral transduction.

---

Hence, our functional data did not support the hypothesis that the inhibitory activity of the accessory helix arises from insertion into the SNARE complex to replace part of the synaptobrevin SNARE motif, in correlation with our NMR and ITC results.

## Inhibition of spontaneous release by the accessory helix of dmCpx

In search of an alternative model that could explain the inhibitory function of the complexin accessory helix, we turned our attention to complexin from *Drosophila melanogaster* (dmCpx) because a dramatic increase in spontaneous release is observed in its absence (*Huntwork and Littleton, 2007*) and experiments with chimeras of murine CpxI and dmCpx suggested that the accessory helix of dmCpx inhibits release more strongly than that of murine CpxI in mouse hippocampal neurons (*Xue et al., 2009*). To verify this conclusion, we generated a chimeric complexin with most of the sequence corresponding to murine CpxI but with the accessory helix of dmCpx (dmAcc-CpxI; *Figure 8A*), and analyzed its influence on neurotransmitter release in complexin triple KO neurons. No significant differences in evoked EPSC amplitudes, RRP charge and Pvr were observed between neurons expressing the dmAcc-CpxI chimera or WT CpxI (*Figure 8B–D*). The PPR analyzed from trains of five EPSCs at 50 Hz also showed no difference between chimeric and WT rescue (*Figure 8E*). At longer stimulations, the short-term plasticity characteristics at 10 Hz revealed a slight, but not significant, decrease in depression when the chimeric dmAcc-CpxI was expressed (*Figure 8F*). Interestingly however, we did observe a significant, ca. 30% reduction in the frequency of spontaneous release in KO neurons expressing dmAcc-CpxI compared to KO neurons expressing WT CpxI, while the amplitudes of the miniature EPSCs (mEPSCs) were not different (*Figure 8G*). These results indicate that the accessory alpha helix of dmCpx contains some feature(s) that renders it more inhibitory than the accessory helix of mammalian CpxI.

## Changing the charge of the accessory alpha helix alters neurotransmitter release

The accessory helix of dmCpx contains only one hydrophobic residue, like mammalian CpxI (*Figure 8A*); therefore, hydrophobicity does not explain the enhanced inhibitory activity of the dmCpx accessory helix. Sequence comparisons suggest that the accessory helix of dmCpx is longer and more negatively charged than that of mammalian CpxI. These observations and the crystal structure of the CpxI(26-83)/SNARE complex (*Figure 1A*) lead naturally to a simple model whereby the complexin accessory helix inhibits release because it is oriented toward the area where the two membranes need to be brought together for fusion and this action is hindered by electrostatic repulsion between the accessory helix and the two membranes (*Figure 1E*).

To test this model, we made two mutants of mammalian CpxI where we changed the charge of the accessory helix, one where we added five negative charges (CpxI-5E) and another where we replaced three negatively charged with positive charges (CpxI-3R) (*Figure 9A*). Lenti-viral expression of the positively charged CpxI-3R and WT CpxI in complexin I-III triple KO neurons did not reveal significant differences in EPSC amplitudes, but rescue with the negatively charged CpxI-5E mutant yielded slightly reduced EPSC amplitudes (*Figure 9B*). The RRP sizes in the four different groups analyzed were not significantly different (*Figure 9C*). Calculation of the Pvr showed similar release probabilities,

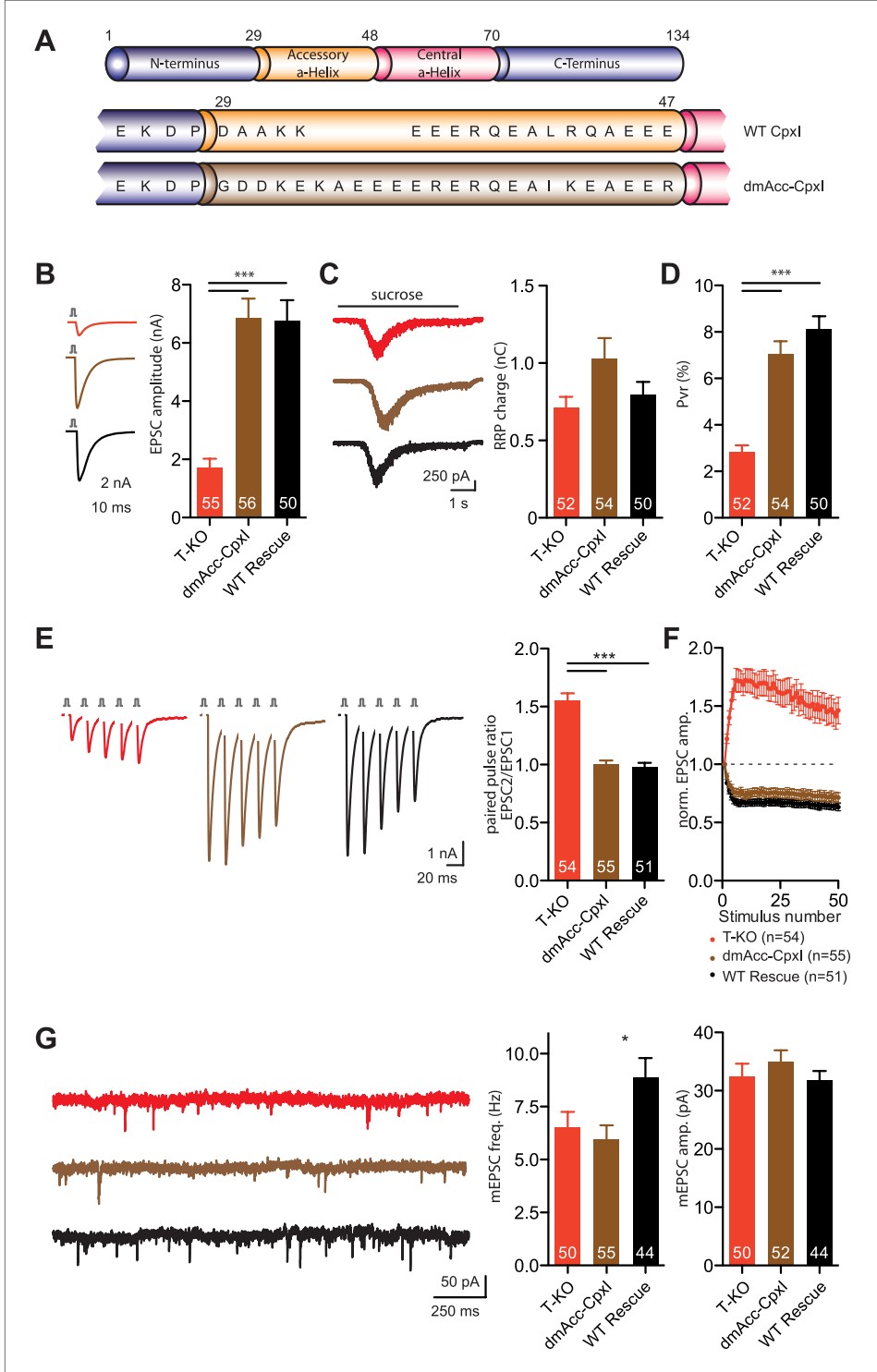

**Figure 8**. Inhibition of spontaneous release by the accessory alpha helix of dmCpx. (**A**) Overview of the replacement of the accessory helix of CpxI with the accessory helix sequence of drosophila Cpx. (**B**) Representative traces and summary data of evoked EPSC (**B**) and synaptic responses to hypertonic sucrose solution (RRP) (**C**) of T-KO, dmAcc-CpxI and WT-CpxI expressing hippocampal neurons. (**D**) Bar graph of the calculated Pvr. (**E**–**F**) Analysis of short-term plasticity behavior: Example traces of a train of 5 APs at 50 Hz of T-KO, dmAcc-CpxI and WT CpxI expressing neurons from which the paired pulse ratio was calculated (**E**) and amplitudes of 50 EPSCs evoked at 10 Hz which were normalized to the first EPSCs and plotted over stimulus number (**F**). (**G**) Spontaneous transmitter release:
*Figure 8. Continued on next page*

*Figure 8. Continued*

Representative traces of T-KO, dmAcc-CpxI and WT-CpxI expressing neurons and summary data of mEPSC frequency and mEPSC amplitude. Data are expressed as mean ± SEM, *p<0.05; ***p<0.001. The numbers of neurons analyzed are shown within the bars.

with a tendency in the CpxI-3R mutant towards higher release probability, and a tendency in the CpxI-5E mutant towards lower probability (*Figure 9D*). Consistent with the Pvr results, trains of five EPSCs at 50 Hz showed that the facilitatory synaptic short-term plasticity characteristic of the complexin I-III triple KO could be rescued in the CpxI-3R expressing neurons to the same level as the WT CpxI expressing neurons (*Figure 9E*). However, expressing the negatively charged CpxI-5E protein did not rescue this KO phenotype to the same extent, and the PPR in CpxI-5E expressing neurons was significantly increased compared to WT CpxI expressing neurons (*Figure 9E*). These rescue behaviors of the different charged versions of CpxI were also observed when analyzing the short-term plasticity characteristics from trains of 50 EPSCs at 10 Hz. In this case, CpxI-3R expressing neurons showed a slight increase in depression whereas CpxI-5E expressing KO neurons did not depress to the same extent as WT CpxI expressing neurons (*Figure 9F*). Importantly, complexin I-III KO neurons expressing the positively charged CpxI-3R exhibited a considerable increase in mEPSC frequency, whereas expression of the negatively charged CpxI-5E resulted in a significant decrease in mEPSC frequency compared to the WT CpxI expressing KO neurons (*Figure 9G*). The amplitudes of these events were not different (*Figure 9G*).

Collectively, these results indicate that the accessory alpha helix of CpxI exerts its inhibitory function at least in part through the presence of negatively charged residues and that changes towards a more positively or more negatively charged nature result in decrease or increase of the inhibitory effect on spontaneous neurotransmitter release, respectively.

### Influence of the complexin accessory helix on the fusogenicity of synaptic vesicles

Absence of complexins does not alter the RRP as defined from the release induced by 500 mM hypertonic sucrose in autaptic hippocampal neurons (e.g., *Figures 7B, 8C and 9B*) but does lead to a decrease in release caused by 250 mM hypertonice sucrose (*Xue et al., 2010*), showing that complexins increase the propensity of synaptic vesicles to fuse. To examine whether changing charged residues in the accessory helix of CpxI affects fusogenicity, we compared the responses to 250 mM sucrose in complexin I-III KO neurons expressing WT CpxI, CpxI-3R or CpxI-5E. Compared to WT CpxI expressing neurons, expression of the less negatively charged mutant, CpxI-3R, led to clear increases in the fraction of the RRP released by 250 mM sucrose and the peak release rate, as well as a decrease in the response onset latency (*Figure 10A–C*). Conversely, introduction of negatively charged residues in the CpxI-5E mutant led to responses to 250 mM sucrose that resembled those of KO neurons, with reduced fractions of RRP release and peak release rates, and increased response onset latencies (*Figure 10A–C*). These data exhibit some correlation with the Pvr (*Figure 10C*) but correlate best with the results from spontaneous release (*Figures 9G, 10C*), suggesting that the charge of the accessory helix has a larger influence on the ability of synaptic vesicles to fuse in the absence of $Ca^{2+}$ than upon $Ca^{2+}$ influx. It seems likely that the stimulatory effects of CpxI and synaptotagmin-1 in evoked release override at least to some extent the inhibition by the CpxI accessory helix.

### Discussion

Complexins are small proteins that play stimulating and inhibitory roles in neurotransmitter release. The inhibitory function was attributed to insertion of the complexin accessory helix into the C-terminus of partially assembled SNARE complexes (*Xue et al., 2007*; *Giraudo et al., 2009*; *Kummel et al., 2011*), but the validity of this model was unclear. In the study presented here, we were unable to detect any interaction between C-terminally-truncated SNARE complexes and the accessory helix of WT CpxI or the CpxI superclamp mutant using highly sensitive biophysical methods in solution. Moreover, we find that the effects of superclamp and poor-clamp CpxI mutations on neurotransmitter release do not correlate with their effects on cell–cell fusion assays, actually pointing in opposite directions. We also show that mutations that increase the negative charge of the accessory helix inhibit neurotransmitter release while mutations that increase its positive charge enhance release. These results

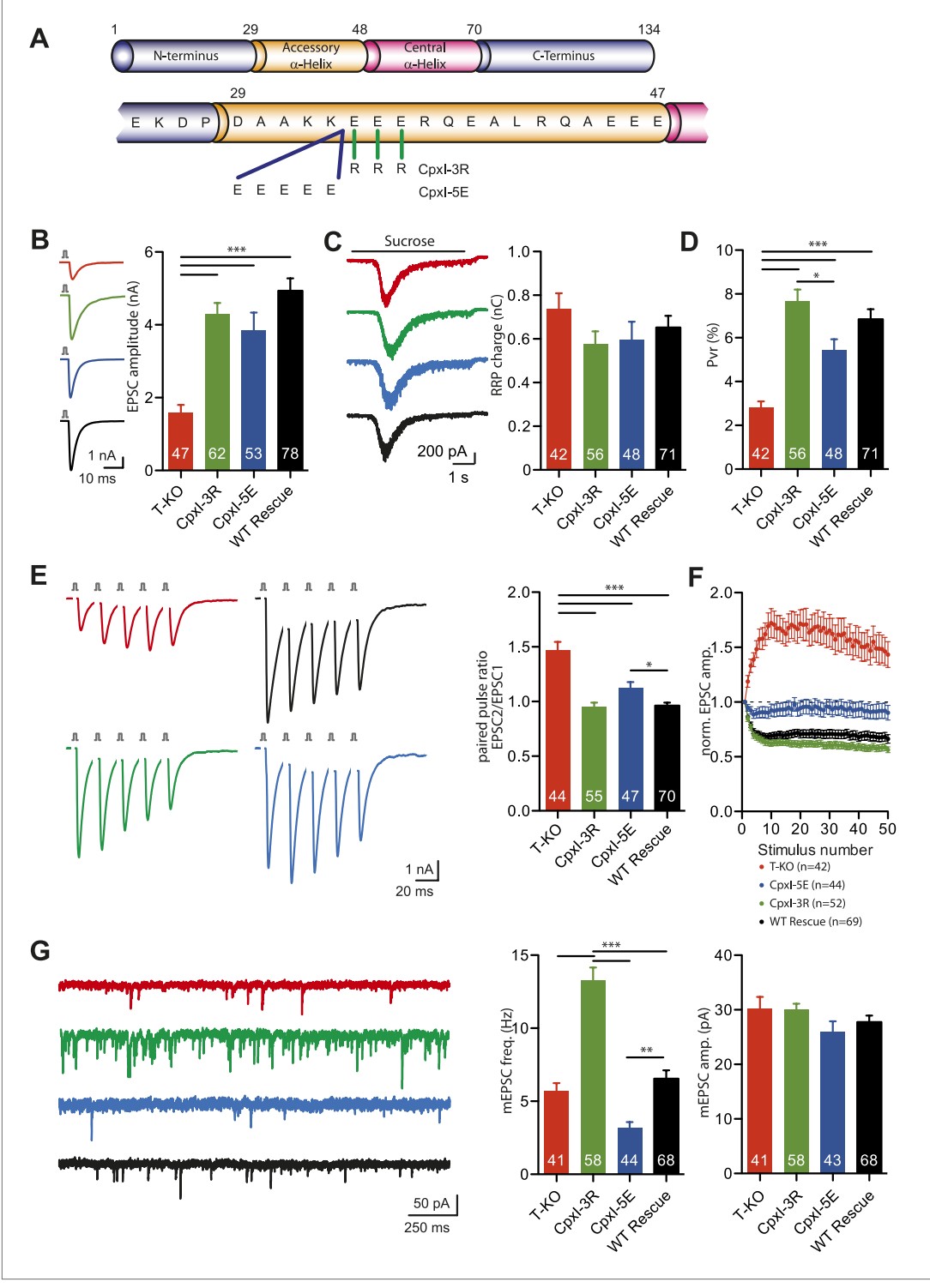

**Figure 9**. Inhibitory effect of the accessory alpha helix is charge dependent. (**A**) Overview of the CpxI accessory alpha helix sequence and the introduced mutations resulting in more positively charged (CpxI-3R) or more negatively charged (CpxI-5E) accessory alpha helix. (**B–C**) Representative traces and summary data of evoked EPSC (**B**) and synaptic responses to hypertonic sucrose solution (RRP) (**C**) of T-KO, CpxI-3R, CpxI-5E and WT-CpxI expressing hippocampal neurons. (**D**) Bar graph of the calculated Pvr. (**E–F**) Analysis of short-term plasticity behavior: example traces of a train of 5 APs at 50 Hz of T-KO, CpxI-3R, CpxI-5E and WT-CpxI expressing neurons from which the paired pulse ratio was calculated (**E**) and amplitudes of 50 EPSCs evoked at 10 Hz which were
*Figure 9. Continued on next page*

*Figure 9. Continued*
normalized to the first EPSCs and plotted over stimulus number (**F**). (**G**) Spontaneous transmitter release:
Representative traces of T-KO, CpxI-3R, CpxI-5E and WT-CpxI expressing neurons and summary data of mEPSC
frequency and mEPSC amplitude. Data are expressed as mean ± SEM, *p<0.05; **p<0.01; ***p<0.001. The
numbers of neurons analyzed are shown within the bars.

strongly argue against the insertion and zigzag models for the inhibitory activity of the complexin
accessory helix and suggest a simple, alternative model whereby the negative charges of the accessory
helix and perhaps steric hindrance repel both membranes, thus hindering membrane fusion and
neurotransmitter release.

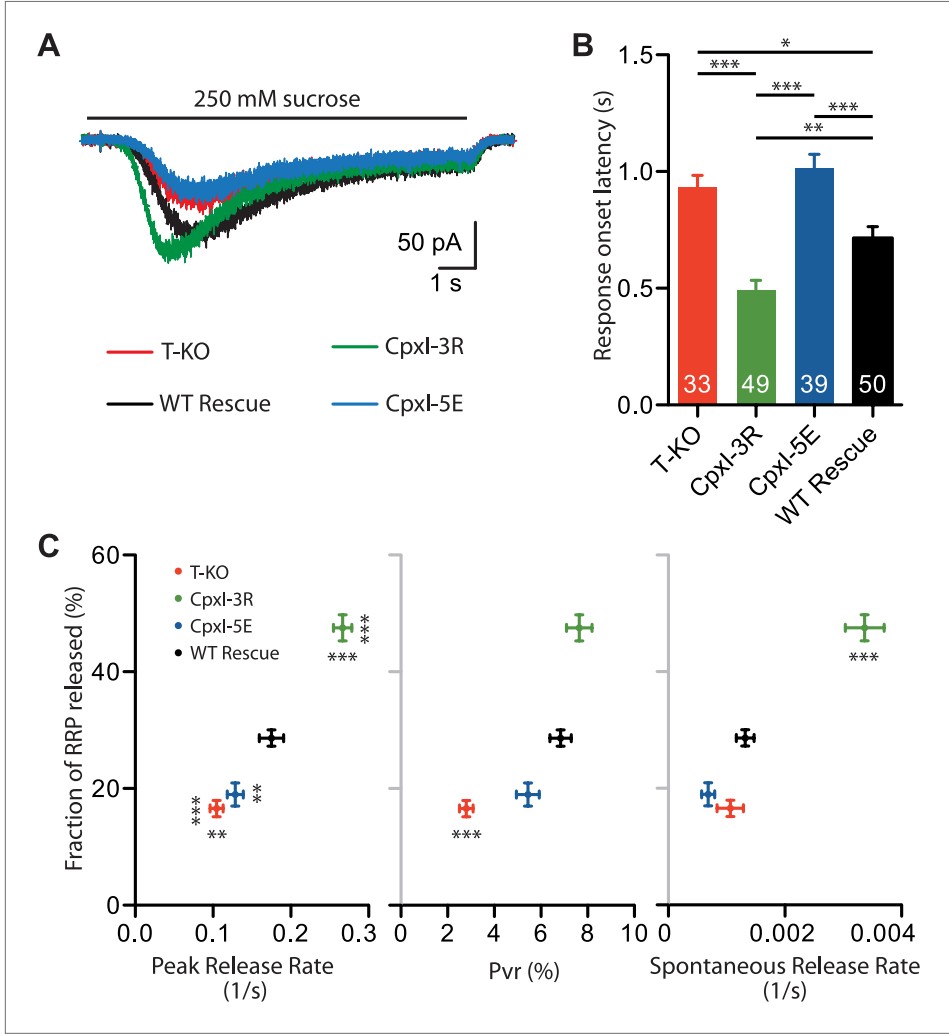

**Figure 10**. Fusogenicity of synaptic vesicles is influenced by the charge of the accessory alpha helix of CpxI.
(**A**) Average traces of synaptic responses induced by 250 mM sucrose solution (T-KO n = 33, CpxI-3R n = 49, CpxI-5E
n = 39, WT CpxI n = 50). (**B**) Summary data of 250 mM sucrose solution-induced response onset latency. The numbers
of neurons analyzed are shown within the bars. (**C**) Correlation plot of fraction of RRP released (T-KO n = 33, CpxI-3R
n = 49, CpxI-5E n = 39, WT CpxI n = 50) vs peak release rate (T-KO n = 33, CpxI-3R n = 49, CpxI-5E n = 39, WT CpxI
n = 50), vesicle release probability (Pvr) (T-KO n = 42, CpxI-3R n = 56, CpxI-5E n = 48, WT CpxI n = 71) and spontane-
ous release rate (T-KO n = 39, CpxI-3R n = 55, CpxI-5E n = 42, WT CpxI n = 63). Data are expressed as mean ± SEM,
*p<0.05; **p<0.01; ***p<0.001. In (**C**) vertically oriented p values correspond to fraction of RRP and horizontally
oriented p values correspond to peak release rate, Pvr and spontaneous release rate compared to WT rescue.

A major question that arises from our study is: Can the insertion and zigzag models be now completely ruled out? In addressing this question, it is critical to consider the available data and the arguments that have been used to support these models:

1. We originally proposed that part of the accessory helix might insert into partially assembled SNARE complexes (*Figure 1C*) to explain the inhibitory function of this helix (*Xue et al., 2007*), but this proposal was not based on any biochemical data and the model could be questioned based on the paucity of hydrophobic residues in the accessory helix.

2. The related model whereby the entire accessory helix inserts into a partially assembled SNARE complex was supported by the effects of superclamp and poor-clamp mutations (e.g., D27L, E34F, R37A, and K26A) in cell–cell fusion assays (*Giraudo et al., 2009*). However, the model envisioned that the three charged side chains replaced in the superclamp mutant (D27, E34, and R37) insert into the hydrophobic groove left in the SNARE complex, which is very unlikely from a thermodynamic point of view. Moreover, our data show that the effects of these mutations on cell–cell fusion do not correlate with their effects on neurotransmitter release: the superclamp D27L, E34F, R37A mutation that enhanced inhibition of cell–cell fusion had little effect on release, with a slight tendency to stimulate release, while the K26A mutation that diminishes the inhibitory acitivity of CpxI in cell–cell fusion actually impairs release (*Figure 7*).

3. It has been argued (e.g., *Kummel et al., 2011*) that the D27L, E34F, R37A superclamp mutation inhibits neurotransmitter release in vivo based on rescue experiments on complexin knockdown neurons (*Yang et al., 2010*). However, these rescue experiments revealed no effect of the D27L, E34F, R37A mutation on evoked release, and it was unclear whether a modest inhibitory effect on spontaneous release was significant. Our rescue data with this mutant using complexin I-III triple KO neurons (*Figure 7*) are consistent with the results of the knockdown rescues considering the small nature of the effects observed, and the experimental differences between the two approaches. Thus, neither of the two studies supports the insertion or zigzag models.

4. The crystal structure of the CpxI(26-83) D27L, E34F, R37A superclamp mutant with the Δ60 synaptobrevin-truncated SNARE complex was purported to demonstrate an alternative insertion model whereby the accessory and central helices bind to different SNARE complexes (zigzag model) (*Kummel et al., 2011*) (*Figure 1B*). However, the binding mode observed for the mutant accessory helix is highly unlikely thermodynamically for WT CpxI for the above mentioned reason that three mutated side chains are charged and hence are unlikely to insert into a hydrophobic groove. Correspondingly, our extensive NMR analyses did not detect any interaction between the WT CpxI accessory helix and synaptobrevin-truncated SNARE complexes in solution (*Figures 2, 3*, *Figure 3—figure supplement 1C*). We were also unable to detect any interaction in solution between truncated SNARE complexes and the accessory helix even for CpxI superclamp mutant fragments (*Figure 2—figure supplement 2*, *Figure 3—figure supplement 1A,D*, *Figure 4E*). A plausible explanation for these findings is that the interaction between the superclamp accessory helix and the truncated SNARE complex observed in the crystals is very weak in solution, and hence could not be observed in our assays, but is stabilized by crystallization. Note that, although the interface area with the SNAREs is larger for the accessory helix (ca. 900 $Å^2$ calculated with PISA; *Krissinel and Henrick, 2007*) than for the central helix (ca. 540 $Å^2$), the atomic B-factors of the residues of the accessory helix in the interface with the SNAREs are much larger than those in the central helix interface, with little electronic density for the side chains of the accessory helix interface (*Figure 1—figure supplement 2*). Interestingly, it has been suggested that motion at a crystal packing interface is intermediate between that of a solvent accessible surface and that of a protein core, even for large interfaces (*Carugo and Argos, 1997*). It is also worth noting that *Kummel et al. (2011)* described another structure obtained with a similar CpxI mutant (D27L, E34M, R37A) where four of the eight crystallographically distinct complexes exhibited the same type of interaction with SCΔ60 observed for the D27L, E34F, R37A CpxI mutant, but the other four complexes had an alternative interaction of the accessory helix with the groove of the truncated SNARE complex where the register was shifted by two helical turns (*Kummel et al., 2011*).

5. A competition assay monitored by ITC was used to support the conclusion that the accessory helix of WT CpxI inserts into SCΔ60 (*Kummel et al., 2011*). We can reproduce these data (*Figure 4D*) but it is clear that the underlying assumption that 1.5 equivalentes of CpxI(47-134) saturate SCΔ60

is incorrect (*Figure 4B,C*) and that these ITC assays do not detect an interaction of the accessory helix with SCΔ60.

6.  Experiments with a surface force apparatus (SFA) were proposed to support the zigzag model (*Li et al., 2011a*). However, the effects caused by WT complexin II in these experiments could be subject to more than one interpretation; for instance, they are compatible with both the zigzag model and the electrostatic hindrance model that we propose here.

7.  The zigzag model was also proposed to be supported by FRET measurements showing that the distances between probes placed on CpxI and SNARE complexes increase on truncation of the synaptobrevin C-terminus (*Krishnakumar et al., 2011*; *Kummel et al., 2011*), consistent with the fact that the accessory helix remains close to the SNAREs in the CpxI(26-83)-SNARE complex structure (*Figure 1A*) but points away from the SNAREs in the structure of the CpxI superclamp mutant bound to SCΔ60 (*Figure 1B*). We believe that the interpretation of the FRET measurements in terms of static structures constitutes an oversimplification because multiple evidence suggests that there is some flexibility in the accessory helix even in the non-truncated complex, including the poor dispersion of the NH cross-peaks of the accessory helix, their sharper line widths compared to cross-peaks of the central helix, the fact that the NH and Cα chemical shifts of the accessory helix change much less than those of the central helix upon SNARE complex binding and the high B factors observed in the crystal structure of the complex (*Chen et al., 2002*). Indeed, such flexibility is also consistent with the distance of 20 Å measured by FRET with probes placed at residue 38 of CpxI and residue 193 of SNAP-25 of the SNARE complex (*Kummel et al., 2011*), since the probes would be expected to be much closer according to the structure of the complex (*Chen et al., 2002*) (the closest distance between the side chains of these residues is 10 Å). Moreover, our NMR data indicate that the truncations in the SNARE complex further increase the flexibility of the accessory helix (*Figure 2—figure supplement 1C,D*) and produce flexibility in the N-terminus of the CpxI central helix (*Figure 2C*) as well as in the C-terminus of SNAP-25 where the FRET donor probe was placed (*Figure 3B*, *Figure 3—figure supplement 1C*). The loss in FRET efficiency observed by Kummel et al. upon truncation of the SNARE complex can be readily explained by all these increases in flexibility and the fact that SCΔ60 lacks key residues of synaptobrevin that contact CpxI without the truncation (*Figure 1—figure supplement 1*).

8.  *Kummel et al. (2011)* concluded that flexibility could not explain the FRET efficiency observed between probes placed at residue 38 of CpxI and residue 193 of SNAP-25 in SCΔ60 because there was no detectable FRET in experiments performed with a CpxI mutant containing a flexible GPGP sequence between the accessory and central helices. However, a small decrease in donor fluorescence was actually observed for this mutant (Figure 4D of *Kummel et al., 2011*) and, based on other measurements shown in Figure 4C and Supplementary Table 3 in Kummel et al., 2011, such decrease would correspond to a distance of ca. 42 Å, just 8 Å longer than the distance measured without the GPGP insertion. Considering that a flexible sequence of four residues can readily span 8 Å, that the insertion of these residues is expected to push away the fluorescence probes, and that the error in such long distances is expected to be rather large because of the low associated FRET efficiencies, the results obtained with the GPGP mutant are not inconsistent with the conclusion that increased flexibility underlies the decreased FRET caused by the synaptobrevin truncation.

We believe that, although the arguments presented above and the overall available data argue strongly against the validity of the insertion and zigzag models, it might be premature to completely rule out these models given the complexity of this system. Note for instance that our NMR and ITC data were obtained with truncated SNARE complexes in solution and hence do not rule out the possibility that the CpxI accessory helix interacts with trans-SNARE complexes partially assembled between membranes. Moreover, while cell–cell fusion assays do not correlate with our electrophysiological data or with the stimulatory function of complexins in release, these assays were crucial to establish the functional interplay between complexins and synaptotagmin-1 (*Giraudo et al., 2006*) and did provide support for the insertion model (*Giraudo et al., 2009*). Hence, we believe that it is advisable to keep an open mind about the insertion or zigzag models, but views considering either of these models proven need to be revised, and alternative models need to be considered.

The model proposed here is attractive because of its simplicity and because it emerges naturally from the realization that the accessory helix of complexins is negatively charged (see Supplementary

figure 1 of *Huntwork and Littleton (2007)*), together with examination of the crystal structure of the WT CpxI(26-83)/SNARE complex. Thus, binding of the central helix to the SNARE complex places the accessory helix right between the membranes at the space where they need to be brought together for fusion (*Figure 1E*), and such action is very likely to be hindered at least to some extent by steric and electrostatic repulsion between the negatively charged membranes and the accessory helix. Note also that the stronger inhibitory activity of the accessory helix of dmCpx, compared to the mammalian CpxI accessory helix (*Figure 8*), cannot arise from increased hydrophobicity, but can be explained by our electrostatic hindrance model. The model is also supported by the inhibition of release caused by the 5E mutation, as well as by the increase in release caused by the 3R mutation (*Figure 9*). Moreover, the impairment in release that we observed for the K26A mutant (*Figure 7*; charge change = −1), and the increase in spontaneous release caused in rescue experiments on complexin knockdown neurons by a K26E, L41K, E47K mutation (charge change = +1) (*Yang et al., 2010*) also correlate in general terms with this model. However, these results need to be interpreted with caution, since it is plausible that the 5E and 3R mutations may alter the helical character of the accessory helix, and the considerable magnitude of the effects caused by the K26A and K26E, L41K, E47K mutations suggest that they do not arise only from changes in overall electrostatic potential. Thus, K26 might be involved in some additional, as yet unidentified interaction. Note that K26 does not really form part of the accessory helix in the CpxI(26-83)/SNARE complex and may be involved in releasing the inhibition caused by the accessory helix (*Xue et al., 2010*).

It is also important to note that the accessory helix does not act on its own and its function depends on the neighboring N-terminus. Our previous studies demonstrate an overall facilitatory effect of complexin for evoked and spontaneous release that requires binding of the complexin N-terminus back onto the SNARE complex (*Xue et al., 2010*). This proposed loop may serve two functions, namely to further stabilize the trans-SNARE complex and to properly position the accessory helix near the putative fusion area to reduce spontaneous release through its electrostatic repulsion effect. Alternatively, it is plausible that the function of the accessory helix may involve interactions with other components of the release machinery. Clearly, more research will be required to further test the validity of our model, ideally using diverse approaches. Such approaches should include challenging biophysical studies involving trans-SNARE complexes formed between two membranes, reconstitution experiments that have already yielded a wealth of information on complexins (e.g., *Schaub et al., 2006*; *Yoon et al., 2008*; *Malsam et al., 2009*, *2012*; *Diao et al., 2012*), and correlations with additional studies of neurotransmitter release in neurons.

## Materials and methods

### Recombinant proteins

Expression vectors and protocols for expression and purification of the following protein fragments in *E. coli* were described previously: rat syntaxin 1A residues 191-253, rat synaptobrevin 2 residues 29-93, human SNAP-25 residues 11-82 and 141-203, full-length rat CpxI and rat CpxI residues 26-83 (*Pabst et al., 2000*; *Chen et al., 2002*). Starting from these vectors and from a vector containing the full-length dmCpx sequence (*Xue et al., 2009*), we used standard recombinant DNA techniques and custom designed primers to generate: C-terminally truncated versions of the synaptobrevin SNARE motif (residues 29-60 [Δ60], residues 29-62 [Δ62], residues 29-68 [Δ68], and residues 29-76 [Δ76]); C-terminally truncated versions of the syntaxin-1 SNARE motif (residues 191-228 [Δ228]; residues 191-232 [Δ232] and residues 191-236 [Δ236]); CpxI fragments corresponding to the CpxI accessory helix (residues 26-47 and 26-48) and the central helix plus C-terminus (residues 47-134); mutant versions of the CpxI(26-83) and CpxI(26-48) fragments; and dmCpx(28-88). Proteins were expressed in *E. coli* and purified as described (*Pabst et al., 2000*; *Chen et al., 2002*). Uniform $^{15}$N- or $^{2}$H,$^{15}$N-labeling was performed by growing *E. coli* BL21(DE3) in minimal medium made with $H_2O$ or $D_2O$ as the solvent, respectively, and using $^{15}$NH$_4$Cl as the sole nitrogen source. Uniform $^{2}$H,$^{15}$N,$^{13}$C-labeling of synaptobrevin(29-93) for triple resonance experiments acquired on the $^{2}$H,$^{15}$N,$^{13}$C-Syb-SCΔ236 complex was accomplished by an analogous procedure including $^{13}$C$_6$-glucose as the sole carbon source.

### SNARE complex assembly

Non-truncated SNARE complexes were formed with SNAP-25 (11-82), SNAP-25 (141-203), syntaxin-1(191-253) and synaptobrevin(29-93). For truncated SNARE complexes, synaptobrevin(29-93)

or syntaxin-1(191-253) were replaced by the appropriate C-terminally truncated fragment. In general, complex assembly was accomplished by incubating a mixture of the purified fragments overnight at 4°C and removing remaining unassembled fragments by concentration–dilution cycles with a 30 kDa cutoff, as described for the non-truncated complex (*Chen et al., 2002*). SDS-PAGE monitored by Coomassie blue staining comparing boiled and non-boiled samples was used to confirm that the complexes were fully formed and the unassembled fragments were removed. For the complexes with the most severe truncations (Δ60, Δ62, Δ228, and Δ232), which are not SDS resistant, full complex formation was monitored by non-denaturing PAGE and Coomassie blue staining.

## NMR spectroscopy

All NMR spectra were obtained at 32°C on Agilent (Santa Clara, CA) INOVA800 or INOVA600 spectrometers equipped with triple resonance cold-probes. $^1H$-$^{15}N$ TROSY-HSQC spectra were acquired with samples that normally contained 25-50 µM $^2H$,$^{15}N$-labeled CpxI fragment or SNARE complex $^2H$,$^{15}N$-labeled at one of the SNARE motifs in the absence or presence of a 1.2–1.5 equivalents of unlabeled SNARE complex (truncated or non-truncated) or unlabeled CpxI fragment. The particular fragments used for each figure are described in the corresponding figure legend. Samples containing isotopically-labeled SNARE complexes and unlabeled complexin fragments were dissolved in 25 mM Tris (pH 7.4) containing 125 mM NaCl and 8% $D_2O$. Samples containing isotopically labeled complexin fragmens and unlabeled SNARE complexes were dissolved in 25 mM HEPES (pH 7.1) containing 125 mM NaCl and 8% $D_2O$. TROSY-HNCA and TROSY-HNCOCA spectra with $^2H$ decoupling (*Yang and Kay, 1999*) were used to obtain partial backbone assignments for $^2H$,$^{15}N$,$^{13}C$-Syb-SCΔ236 complex as described (*Chen et al., 2002*). All the data were processed with NMRPipe (*Delaglio et al., 1995*) and analyzed with NMRView (*Johnson and Blevins, 1994*).

## Isothermal titration calorimetry

ITC experiments were performed using a VP-ITC system (MicroCal; Northampton, MA) at 37°C in PBS buffer (10 mM $Na_2HPO_4$, 2 mM $K_2HPO_4$ pH 7.4, 2.7 mM KCl, 137 mM NaCl) containing 0.25 mM TCEP. For *Figure 4A,B*, CpxI(47-134) (200 µM) was directly titrated into the chamber containing non-truncated SNARE complex or SCΔ60 (10-15 µM). For *Figure 4C–E*, 200 µM CpxI(47-134), CpxI(26-83) or CpxI(26-83) superclamp mutant (D27L, E34F, R37A) were titrated in the chamber containing 10-15 µM SCΔ60 and 1.5 equivalents of CpxI(47-134). All proteins were dialyzed in the same buffer before the experiments. The data were fitted with a nonlinear least squares routine using a single-site binding model with Origin for ITC v.5.0 (Microcal).

## Lentiviral constructs and virus production

For expression of CpxI variants within neuronal cells a modified lentiviral vector (*Lois et al., 2002*) was used in which a human *Synapsin-1* promoter, driving the expression of CpxI, and a second promoter (ubiquitin C), which serves as driver for the reporter gene (EGFP), were employed. WT rat CpxI (GenBank accession number: NM_022864) and *Drosophila* Cpx (AY121629) cDNAs were used to generate all Cpx variants by standard recombinant DNA techniques. For immunocytochemistry a 3xFLAG epitope (Sigma-Aldrich) was fused at the C-terminus of CpxI. After sequence verification, the cDNAs were cloned into the lentiviral shuttle vector and lentiviral particles were prepared as described (*Lois et al., 2002*). Briefly, HEK293T cells were cotransfected with 10 µg shuttle vector and the helper plasmids pCMVdR8.9 and pVSV.G (5 µg each) with X-tremeGENE 9 DNA transfection reagent (Roche Diagnostic). After 72 hr the virus containing cell culture supernatant was collected and purified by filtration. Aliquots were flash-frozen in liquid nitrogen and stored at −80°C. Viruses were titrated with WT hippocampal mass-cultured neurons. For infection, about $5 \times 10^5$–$1 \times 10^6$ infectious virus units were pipetted onto 1 DIV hippocampal CpxI-III triple KO neurons per 35 mm-diameter well.

## Neuronal culture

Murine microisland cultures were prepared as described (*Xue et al., 2007*). CpxI-III triple KO neurons were described previously (*Xue et al., 2008b*). Animals were handled according to the rules of Berlin authorities and the animal welfare committee of the Charité Berlin, Germany. Primary hippocampal neurons were prepared from mice on embryonic day E18 and plated at 300 cm$^{-2}$ density on WT astrocyte microisland for autaptic neuron electrophysiology. For western blotting and immunocytochemistry

hippocampal neurons were plated at 10.000 cm$^{-2}$ and 5000 cm$^{-2}$, respectively, on continental WT astrocyte feeder layer.

## Electrophysiology of cultured neurons

Whole cell patch-clamp recordings in autaptic neurons were performed as previously described (*Xue et al., 2009*). The extracellular solution contained (in mM) 140 NaCl, 2.4 KCl, 10 Hepes, 2 CaCl$_2$, 4 MgCl$_2$, 10 Glucose (pH adjusted to 7.3 with NaOH, 300 mOsm). The patch pipette solution contained (in mM) 136 KCl, 17.8 Hepes, 1 EGTA, 0.6 MgCl$_2$, 4 ATP-Mg, 0.3 GTP-Na, 12 phosphocreatine and 50 units/ml phosphocreatine kinase (300mOsm, pH 7.4). Neurons were clamped at −70 mV with a Multiclamp 700B amplifier (Molecular Devices; Sunnyvale, CA) under control of Clampex 9 (Molecular Devices) at DIV 11-17. Data were analyzed offline using Axograph X (AxoGraph Scientific; Berkeley, CA) and Prism 5 (GraphPad Software; La Jolla, CA). Statistic significances were tested using one-way analysis of variance followed by a Tukey post hoc test to compare all groups.

EPSCs were evoked by a brief 2 ms somatic depolarization to 0 mV. EPSC amplitude was determined as the average of 5 EPSCs at 0.1 Hz. RRP size was determined by measuring the charge transfer of the transient synaptic current induced by a pulsed 5 s application of hypertonic solution (500 mM sucrose in extracellular solution). Pvr was calculated as the ratio of the charge from an evoked EPSC and the RRP size of the same neuron. Evoking 5 or 50 synaptic responses at 50 or 10 Hz respectively in standard external solution analyzed short-term plasticity. PPR was calculated by dividing the second EPSC amplitude with the first EPSC amplitude from the average of three 50 Hz trains at 0.1 Hz. For analyzing mEPSCs, traces were digitally filtered at 1 kHz offline. Then the last 8 s of 5 traces of EPSCs at 0.1 Hz were analyzed using the template-based mEPSC detection algorithm implemented in Axograph X (AxoGraph Scientific) and substracted from background noise by detecting events in the last 3 s of 5 EPSCs at 0.2 Hz in 3 mM kynurenic acid in extracellular solution.

Synaptic-vesicle fusogenicity was measured by applying 250 mM sucrose solution onto the neuron for 10 s and analyzed as described previously (*Xue et al., 2010*). Briefly, to obtain the fraction of RRP released at 250 mM sucrose solution, the charge transfer of the transient synaptic current was measured and divided by the RRP size obtained by 500 mM sucrose application (5 s) from the same neuron. The response onset latency was calculated between the open tip control for solution exchange and the onset of the sucrose response. The peak release rate was calculated by dividing peak amplitude of sucrose response with the RRP size of the same neuron. The spontaneous release rate was calculated by dividing the mEPSC frequency with the number of vesicles within the RRP. This number was obtained by multiplying the mEPSC charge with the RRP charge measured by 500 mM sucrose application.

## Western blotting and immunocytochemistry

For detection of CpxI protein levels by western blotting, protein lysates were obtained from mass cultures of CpxI-III KO hippocampal neurons (DIV 14) grown on WT astrocyte feeder layers. Briefly, cells were lysed using 50 mM Tris/HCl (pH 7.9), 150 mM NaCl, 5 mM EDTA, 1% Triton-X-100, 1% Nonidet P-40, 1% sodium deoxycholate, and protease inhibitors (complete protease inhibitor cocktail tablet, Roche Diagnostics GmbH; Manheim, Germany). Proteins were separated by SDS-PAGE and transferred to nitrocellulose membranes. After blocking with 5% milk powder (Carl Roth GmbH) for 1 hr at room temperature, membranes were incubated with rabbit anti-CpxI/II (1:1000; Synaptic System) and mouse anti-tubulinIII (1:750; Sigma–Aldrich) antibodies overnight at 4°C. After washing and incubation with corresponding horseradish peroxidase-conjugated goat secondary antibodies (all from Jackson ImmunoResearch Laboratories), protein expression levels were visualized with ECL Plus Western Blotting Detection Reagents (GE Healthcare Biosciences).

To detect synaptic localization by immunocytochemistry, lentiviral transduced neurons were washed once in PBS, fixed in 4% paraformaldehyde for 10 min at room temperature and treated 3 times 5 min with 100 mM glycine in PBS. Then cells were blocked with 5% normal goat serum and 0.1% Tween-20 in PBS for 1 hr and incubated with primary antibodies overnight at 4°C in blocking solution. The following antibodies were used: mouse anti-FLAG (1:500; Sigma-Aldrich; Saint Louis, MO), guinea pig anti-VGlut1 (1:4000; Synaptic System). Primary antibodies were labeled with anti-mouse Rhodamine Red and anti-guinea pig Alexa Fluor 405 (each 1:500; Jackson Immunoresearch Laboratories; West Groove, PA) for 1 hr at room temperature. After washing, cover slips were mounted with Mowiol 4-88

antifade medium (Polysciences Europe GmbH; Eppelheim, Germany). Neurons were imaged using an Olympus IX81 microscope.

## Acknowledgements

We thank Annegret Felies, Berit Soehl-Kielczynski, Sabine Lenz, Katja Poetschke, Carola Schweynoch and Bettina Brokowski for technical support, members of the Rosenmund laboratory for discussions, Daeho Lee for initial experiments in this project and Yilun Sun for expert technical assistance on protein expression and purification. We also thank the reviewers of the manuscript who have helped to considerably improve the quality of the paper with their constructive criticisms. This work was supported by the German Research Council (DFG Collaborative Research Grant SFB665, 958 to TT and CR), the European Research Foundation (Grant SynVglut to CR), the Excellence Cluster Neurocure Exc257 (to CR), the Welch foundation (grant I-1304 to JR) and the NIH (grant NS37200 to JR).

## Additional information

### Competing interests

CR: Reviewing editor, *eLife*. The other authors declare that no competing interests exist.

### Funding

| Funder | Grant reference number | Author |
| --- | --- | --- |
| Deutsche Forschungsgemeinschaft | SFB665, 958 | Thorsten Trimbuch, Christian Rosenmund |
| European Research Council | SynVglut | Christian Rosenmund |
| Excellence Cluster Neurocure | Exc257 | Christian Rosenmund |
| Welch Foundation | I-1304 | Josep Rizo |
| National Institutes of Health | NS37200 | Josep Rizo |

The funders had no role in study design, data collection and interpretation, or the decision to submit the work for publication.

### Author contributions

TT, JR, CR, Conception and design, Acquisition of data, Analysis and interpretation of data, Drafting or revising the article; JX, DRT, Conception and design, Acquisition of data, Analysis and interpretation of data; DF, Acquisition of data, Analysis and interpretation of data

### Ethics

Animal experimentation: Animals were handled according to the animal welfare committee of the Charité-Universitätsmedizin Berlin, Germany. Time pregnant females were anesthetized and euthanized at E18 according to the german animal law (§4 section 1) and permit by the Berlin authorities (Landesamt für Gesundheit und Soziales, LaGeSo) under the permit number T0220/09.

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
