## [Decision Letter]

Thank you for sending your work entitled “An electrostatic hindrance model for accessory helix-mediated inhibition of neurotransmitter release by complexins” for consideration at *eLife*. Your article has been generally favorably evaluated by a Senior editor and 3 reviewers, one of whom is a member of our Board of Reviewing Editors, although there were some concerns.

The following individuals responsible for the peer review of your submission have agreed to reveal their identity: Axel Brunger (Reviewing editor); William Wickner and Ad Bax (peer reviewers).

The Reviewing editor and the other reviewers discussed their comments before we reached this decision, and the Reviewing editor has assembled the following comments to help you prepare a revised submission.

This work questions the interaction between the accessory helix and a truncated ternary SNARE complex. In 2011, multiple experiments by the Rothman, Reinisch, and Tareste groups published: ITC binding experiments, crystal structures with a mutant of complexin in complex with a truncated SNARE complex, solution (bulk) FRET data, and surface force measurements. However, among these four different experiments, the only direct demonstrations of an interaction between the accessory helix of complexin and the SNARE complex were the crystal structures, and the ITC experiment.

In this work, it is shown that the interpretation of ITC experiment may have suffered from an incorrect assumption of the affinity between the complexin and SNARE fragments used, and that the data can be explained in an alternative fashion that does not involve an interaction between the accessory helix of complexin and the truncated SNARE complex. Rizo, Rosenmund and co-workers then present a large body of solution NMR data that does not show any evidence for an interaction between the accessory helix and the truncated SNARE complex, even for the so-called superclamp mutant, suggesting the interaction represents an extremely weak, lowly occupied state. In addition, the authors argue that the bulk FRET data by Kummel et al. can be interpreted in an alternative fashion (but this re-interpretation is questionable, see below).

These in vitro studies are augmented by experiments with cultured neurons that show that mutation of the accessory helix modulates release based on the charge distribution on the accessory helix. The more negative, the more fusion is inhibited. It is suggested that the inhibitory function of complexin is caused by an interaction between the membrane and the accessory helix. Moreover, they show that 'super'-clamp (D27L,E34F,R37A) and a 'no'-clamp (K26A) mutants of complexin do not have the expected effects when introduced into actual neurons.

That being said, support for the newly introduced electrostatic repulsion idea is rather thin and is contradicted by other data (see below). It is even is unlikely that the accessory helix would adopt a helical conformation in the presence of the rather severe mutations of Figure 9. Considering the significant impact of the CpxI K26A mutation, it's more likely that the CpxI role is more subtle, along the lines suggested at the end of the Discussion section.

Regardless if the electrostatic repulsion model is correct (it will certainly require a lot of discussion, or perhaps be removed, in a revised version of this paper), the main point of this paper is that it entirely re-opens the question how complexin really works.

Issues to address:

1) Although the overlaid TROSY-HSQC spectra are reasonably compelling for the general argument the authors are trying to make, it would be helpful if more specific interpretation were supported by an SI figure showing the (composite 1H/15N) amide shift changes as a function of residue number for the delta62 and delta68 truncations. Similarly, a correlation graph of CpxI shift changes (vs. full length minus random coil) for the shift changes in the N-terminus of the central helix would provide stronger support for the “increasingly flexible toward the N-terminus” statement. Alternatively, 15N relaxation measurements would make this more quantitative, but not necessary from my perspective if indeed the shift change is in the random coil direction.

2) The negative charge repulsion model is contradicted by the following experiments: the non-clamping mutants by Kummel et al. (L41E, A44E, A30E, A31E) and (A30E and A31E) are both negatively charged, but yet, they have no effect on spontaneous fusion in their cell-based fusion assay. The poor-clamp mutant (K26E, K41K, E47K, also somewhat negatively charged) by Yang et al. (Neuron, 2010) actually increases spontaneous release in neuronal cultures. These contradictions need to be addressed by additional experiments, or the negative charge model needs to be modified.

3) As mentioned in the Discussion, point 4, Reinish et al. determined a second crystal form (with an additional mutation in complexin, F34M, in addition to the “superclamp” mutations in the accessory helix) where the accessory helix interacts in two different registers. This indicates promiscuity of the interaction between the accessory helix and the 2nd SNARE complex in the crystal structures. This point needs some clarification in the Discussion: first these two alternative forms only occur upon the additional F34M mutation. Second, it is not necessarily that crystallization has induced this interaction. It is possible that this weak interaction is a lowly state that is too infrequent to be observed by NMR, but has been stabilized in the context of the crystal structure.

4) The interaction between the accessory helix and a neighboring partially folded full SNARE complex is not ruled out by the presented experiments since they do not involve partially folded full SNARE complex. This should be noted in the paper.

5) The alternative explanation of the original Kummel et al., ITC data and the additional ITC experiments are convincing. However, these data do not rule out an interaction between the accessory helix and SNARE complex per se, but it may be much weaker than estimated by the original Kummel et al. experiment. It would be prudent to add this caveat to the discussion of the ITC experiments. Did the authors attempt to conduct the competition experiment under actually saturating conditions of CpxI(47-134)?

6) Kummel et al. used bulk fret and measured the FRET between S25 Cterm (residue 193) and CX Nterm (residue 38). They showed that the more VAMP is truncated at the C-terminus, truncation of VAMP up to residue 60, the weaker the FRET efficiency. In this work, the NMR results and the weaker affinity observed by ITC suggests that the VAMP truncations make SNAP-25 and also syntaxin more flexible and unstructured, thus destabilizing the interaction with complexin as well, resulting in lower FRET. Thus, an interpretation of the NMR result is that the lower FRET efficiency is a result of motional averaging of multiple conformations that produces the appearance of a lower FRET “state”. However, Kummel et al. did an experiment to rule out this possibility: They used a 'flexible' CPX construct (CPX-GPGP), in which a helix-breaking GPGP linker was inserted between the central and accessory helices of CPX (Figure 4, Supplemental Table 3 in Kummel et al.). To quote Kummel et al. “these experiments indicate that when CPX is bound to a half-zippered form of the SNARE complex, CPXacc has the intrinsic property of extending away from the complex. Because this conformation is maintained in solution, it determines how the complex crystallizes and not vice versa.”

[Editors' note: further clarifications were requested prior to acceptance, as described below.]

Thank you for sending your work entitled “An Electrostatic-Steric Hindrance Model for Accessory Helix Mediated Inhibition of Neurotransmitter Release by Complexin” for further consideration at *eLife*. Your article has been favorably evaluated by a Senior editor and 3 reviewers, one of whom is a member of our Board of Reviewing Editors.

The following individuals responsible for the peer review of your submission have agreed to reveal their identity: Axel Brunger (Reviewing editor); William Wickner and Ad Bax (peer reviewers).

The Reviewing editor and the other reviewers discussed their comments before we reached this decision, and the Reviewing editor has assembled the following comments to help you prepare a revised submission.

We thank the authors for addressing some of our concerns and would be ready to recommend acceptance after addressing the following remaining concerns:

1) We are disappointed that the authors did not address the question if the charge-altering mutations of the accessory helix retain the helical character of the accessory helix (suggested by the strong alpha-helical chemical shifts of the accessory helix by [38] J.Biol.Chem). It is certainly possible that this happens, but the data presented here do not provide direct (e.g., NMR) evidence for this notion. Moreover, the proposed model of electrostatic-steric hindrance by the accessory helix remains speculative. It is of course fine to provide such a speculative model in the Discussion section of the paper. However, the title is misleading since it suggests that the model has been definitely proven in this work. Rather, the main focus of this paper is a re-examination of the interaction between the accessory helix and the truncated SNARE complex that was previously studied by Kummel et al. Thus, we request a change of title to properly reflect the main focus of this work.

2) The statement in the Results section: “these cross-peaks actually increased in the spectrum of Cpx(26-83) bound to SCdelta68 with respect to the SC-bound state, and increased somewhat more in the SCdelta62-bound spectrum (Figure 2—figure supplement 1) These data show that the synaptobrevin C-terminal truncations increase the flexibility of the accessory helix, in correlation with the destabilization of N terminal part of the central helix,”. The best resolved accessory helix resonances (A30 and Q38) actually do not seem to increase from SCdelta68 to SCdelta62. Thus, the statements about the increase in flexibility of the accessory helix need to be qualified.

3) The discussion of the Kummel at al. crystal structure (point 4 in the Discussion) remains a concern. Firstly, the area of the interface between the accessory helix and the truncated SNARE complex is actually quite reasonable (715 A^2) and this in itself does not suggest that this contact is induced by “crystal packing”. Secondly, the last statement “that the interactions leading to the zigzag array are not specific and are induced by crystallization” implies that crystal packing may induce non-specific interactions. To the contrary, the observation of such an interaction in different crystal forms and for different mutations of residue 34 (albeit with a register shift observed in the second crystal form for some of the non-crystallographic related molecules) actually speaks against an entirely non-specific interaction. Moreover, the crystallization conditions seem not unusual, and the pH is comparable for the crystallization and NMR experiments. There is one clear difference: in the crystal structure, two complexin molecules are interacting with the truncated SNARE complex. It is theoretically possible that binding of the central complexin helix to the SNARE complex is required in order to enable binding of the accessory helix of the second complexin molecule. Thus, we remain concerned about this difference between the crystal structures and the solution NMR studies on the same system, and the discussion must be modified to properly describe this discrepancy, and provide possible alternative explanations.

4) The authors may wish to explicitly leave wiggle room for future discovery of some such an interaction, e.g., with a sub-complex of SNAREs. For example, one could write “While current data do not support the proposed zigzag SNARE:complexin interaction, it remains possible that such an interaction might occur under some biological condition, or with some sub-complex of SNAREs.”

---

## [Author Response]

Thank you for the nice summary of our work and its significance. We agree that the main point of the paper is that it entirely opens the question of how complexin inhibits release, and that the new model we propose is speculative. However, we still would like to propose the model because it provides a testable hypothesis, because it led to the design of the experiments with the 5E and 3R mutants (otherwise we have no rationale for this design), and because it is supported by the electrophysiological data obtained with these mutants (see also response to point 2). Nevertheless, we present the model as a suggestion and make it clear in the discussion that the model will need to be tested with further research. Moreover, we included the term ‘steric’ in the model (in the title, abstract, etc.) because the hindrance caused by the accessory helix could arise from both electrostatic and steric effects, and because this gives further flexibility to the model for testing in the future.

With respect to the comment ‘*It is even is unlikely that the accessory helix would adopt a helical conformation in the presence of the rather severe mutations of*
Figure 9’, glutamate and arginine are generally compatible with helical conformation, and secondary structure predictions with Jpred still predict helical conformation for the accessory helix of the 3R and 5E mutants (see below). In addition, consecutive residues with the same charge occur to some extent in the native accessory helices of mammalian and dm CpxI (Figure 8). Importantly, the 3R and 5E mutants are largely rescuing the phenotype of the complexin I-III triple KO (Figure 9). Therefore, although it is plausible that the large number of consecutive positive or negative charges in the mutants might destabilize the helix, the mutations do not cause a major impairment of the overall function of complexins.Author response image 1.

*Issues to*
*address:*

*1) Although the overlaid TROSY-HSQC spectra are reasonably compelling for the general argument the authors are trying to make, it would be helpful if more specific interpretation were supported by an SI figure showing the (composite 1H/15N) amide shift changes as a function of residue number for the delta62 and delta68 truncations. Similarly, a correlation graph of CpxI shift changes (vs. full length minus random coil) for the shift changes in the N-terminus of the central helix would provide stronger support for the “increasingly flexible toward the N-terminus” statement. Alternatively, 15N relaxation measurements would make this more quantitative, but not necessary from my perspective if indeed the shift change is in the random coil direction*.

We agree with the reviewers that it is desirable to provide a more detailed analysis of these data, with some quantification that supports our conclusions. Such detailed analysis is hindered by the fact that multiple cross-peaks of the Cpx(26-83) central helix cannot be unambiguously identified upon truncation of the SNARE complex, but the data for the cross-peaks that can be identified are still informative. In Figure 2—figure supplement 1, we now present plots of chemical shift changes caused by truncation of synaptobrevin to residue 68 compared to the changes observed when comparing free and SNARE complex-bound Cpx(26-83).

In one plot we use the latter changes for normalization to provide a measure of how much the central helix is destabilized, while in the second plot we correlate the first changes with the latter. Based on these plots, we have changed slightly the interpretation, suggesting that ‘the N-terminal half of the accessory helix becomes flexible’ rather than ‘becomes increasingly flexible toward the N-terminus’ Note that we prefer to use the chemical shifts of free Cpx(26-83) rather than random coil chemical shifts because we believe that they provide a better measure of the shifts expected when the central helix is detached from the SNARE complex. Using random coil chemical shifts for the plot presented in Figure 2—figure supplement 1, would yield the plot shown below and lead to analogous conclusions. We would be glad to use this plot instead if the reviewer believes that this is a better choice.Author response image 2.

We have not performed relaxation experiments, but the cross-peak intensities do provide some information on relaxation properties and we now show plots of cross-peak intensities for Cpx(26-83) bound to SC 68 and SC 62, normalized with those of Cpx(26-83) bound to the SNARE complex (Figure 2—figure supplement 1). These plots show that the intensities of the cross-peaks that can be identified from the accessory helix become stronger upon truncation of the synaptobrevin C- terminus, which provides very strong evidence against insertion of the accessory helix into the truncated complex.

*2) The negative charge repulsion model is contradicted by the following experiments: the non-clamping mutants by Kummel et al. (L41E, A44E, A30E, A31E) and (A30E and A31E) are both negatively charged, but yet, they have no effect on spontaneous fusion in their cell-based fusion assay. The poor-clamp mutant (K26E, K41K, E47K, also somewhat negatively charged) by Yang et al. (Neuron, 2010) actually increases spontaneous release in neuronal cultures. These contradictions need to be addressed by additional experiments, or the negative charge model needs to be modified*.

At present, there is no clear correlation between the effects of mutations in the cell-cell fusion assay and any physiological data. On the contrary, our electrophysiological data with the superclamp and K26A mutants go in opposite directions from those observed in cell-cell fusion assays. We do not think that this lack of correlation completely rules out the significance of the results obtained with these assays, since at the moment we do not understand the source for the lack of correlation. At the same time, we do not believe that the model emerging from our electrophysiological data and reasonable structural-biophysical arguments should be discarded because of the results obtained with the non-clamping mutants in the cell -cell fusion assays of Kummel et al.

With regard to the K26E, L41K, E47K studied in [61] (please note that residue 41 of WT CpxI is L), the mutation increases the positive charge by 1 unit and increases spontaneous release. This result in principle correlates with our model, although the increase in spontaneous release is rather large. Overall, we feel that the available electrophysiological data with this mutant described in [61], our 3R and 5E mutants, and our mutant containing the dmCpxI accessory helix provide support for the electrostatic model, even though we agree that the model is not proven (and this is how we present it in the paper).

*3) As mentioned in the Discussion, point 4, Reinish et al. determined a second crystal form (with an additional mutation in complexin, F34M, in addition to the “superclamp” mutations in the accessory helix) where the accessory helix interacts in two different registers. This indicates promiscuity of the interaction between the accessory helix and the 2nd SNARE complex in the crystal structures. This point needs some clarification in the Discussion: first these two alternative forms only occur upon the additional F34M mutation. Second, it is not necessarily that crystallization has induced this interaction. It is possible that this weak interaction is a lowly state that is too infrequent to be observed by NMR, but has been stabilized in the context of the crystal structure*.

We would like to clarify that F34M does not really constitute an additional mutation of the WT CpxI sequence but rather replaces one of the side chains that was already mutated in the superclamp mutant (it is D27L,E34M,R37A instead of D27L,E34F,R37A). In the discussion we have tried to explain better what is observed in the crystals of the D27L,E34M,R37A mutant, i.e. two types of interactions in crystallographically distinct complexes, one that is analogous to that observed for the D27L,E34F,R37A mutant and another that has a shift of two helical turns between the accessory helix and the groove of SC 60. We also point out in several parts of the manuscript that the interaction is not observed by NMR but could exist and is not detected because it is very weak.

*4) The interaction between the accessory helix and a neighboring partially folded full SNARE complex is not ruled out by the presented experiments since they do not involve partially folded full SNARE complex. This should be noted in the paper*.

We agree and have included this suggestion in the Discussion.

*5) The alternative explanation of the original Kummel et al., ITC data and the additional ITC experiments are convincing. However, these data do not rule out an interaction between the accessory helix and SNARE complex per se, but it may be much weaker than estimated by the original Kummel et al. experiment. It would be prudent to add this caveat to the discussion of the ITC experiments. Did the authors attempt to conduct the competition experiment under actually*
*saturating conditions of CpxI(47-134)?*

We have performed additional ITC experiments where Cpx(26-83) was titrated over a sample of SC 60 containing 3.0 equivalents of CpxI(1-134) to better saturate the binding site. As expected, only a very small amount of heat was observed in these titrations and this heat corresponds to a natural extension of a direct titration that had reached a molar ratio of 3.0 (new Figure 4—figure supplement 1). Note that, given the sensitivity that we observe in these experiments, more full saturation that would completely eliminate any detectable heat would require a large excess of Cpx(47-134) that could promote aggregation. We believe that the overall data clearly show that no interaction of the Cpx(26-83) accessory helix with SC 60 can be detected in these ITC experiments, and this is how we now present our Conclusion.

*6) Kummel et al. used bulk fret and measured the FRET between S25 Cterm (residue 193) and CX Nterm (residue 38). They showed that the more VAMP is truncated at the C-terminus, truncation of VAMP up to residue 60, the weaker the FRET efficiency. In this work, the NMR results and the weaker affinity observed by ITC suggests that the VAMP truncations make SNAP-25 and also syntaxin more flexible and unstructured, thus destabilizing the interaction with complexin as well, resulting in lower FRET. Thus, an interpretation of the NMR result is that the lower FRET efficiency is a result of motional averaging of multiple conformations that produces the appearance of a lower FRET “state”. However, Kummel et al. did an experiment to rule out this possibility: They used a 'flexible' CPX construct (CPX-GPGP), in which a helix-breaking GPGP linker was inserted between the central and accessory helices of CPX (*Figure 4*, Supplemental Table 3 in Kummel et al.). To quote Kummel et al. “these experiments indicate that when CPX is bound to a half-zippered form of the SNARE complex, CPXacc has the intrinsic property of extending away from the complex. Because this conformation is maintained in solution, it determines how the complex crystallizes*
*and not vice versa.”*

We believe that the FRET data of Kummel et al. need to be reinterpreted for the following reasons:

A. As we explain in points 7 and 8 of the discussion, our NMR data indicate that the synaptobrevin truncation causes increased flexibility in the CpxI accessory helix and also induces flexibility in the CpxI central helix and SNAP-25 C-terminus, which are well structured without the truncation. Such increases in flexibility can indeed explain the decreased FRET observed upon synaptobrevin truncation in [27].

B. The R0 for the FRET pair used in Kummel et al. was calculated to be 27.5 Å. Hence, they observed good FRET efficiency for distances measured between CpxI and the non-truncated SNARE complex, which were on the order of 20-27 Å. However, the FRET efficiency decreases considerably for the longer distances measured using the truncated SNARE complex (34-42 Å) (values taken from Supplementary Table 3 of Kummel et al.). Hence, it is much more difficult to obtain reliable distances and strong conclusions for these data.

C. Regardless of the issues mentioned in points A and B, the most important point arises from the interpretation of the data obtained with the flexible CPX-GPGP construct mentioned by the reviewer. Kummel et al. concluded that there was no detectable FRET signal for CPX-GPGP (labeled at residue 38) bound to either the truncated or non-truncated SNARE complex, referring to the data shown in Figure 4. However, inspection of Figure 4 shows that there was actually a small decrease in the donor fluorescence in the experiments performed with CPX-GPGP, compared to the donor only control. While this decrease is small, it is comparable to that observed for CPX with the acceptor on residue 31 bound to the truncated SNARE complex (cyan curve in Figure 4), which was used in Supplementary Table 3 to calculate a distance of 42 Å. Note also that the FRET efficiency observed for Cpx labeled at residue 38 (without the flexible GPGP insertion) and bound to the truncated SNARE complex was rather low and yielded a distance of 34 Å (Figure 4 and Supplementary Table 3 of Kummel et al.). Hence, the difference in the distances that can be calculated from the FRET data presented for CPX and CPX-GPGP bound to the truncated SNARE complex is just 8 Å, a rather short distance considering the large errors expected for such low FRET efficiencies. Moreover, the insertion of the GPGP sequence between the central and accessory helices is expected to push the fluorescence probe placed in the Cpx accessory helix away from the probe on the truncated SNARE complex, and it is reasonable for a flexible sequence of four residues such as GPGP to span a distance of approximately 8 Å. Hence, the experiments performed with GPGP do not really rule out the flexibility argument.

In the revised paper we have expanded the discussion of the FRET data to explain these points, while trying to shorten the text in other parts to limit the length of the already long discussion.

[Editors' note: further clarifications were requested prior to acceptance, as described below.]

*1) We are disappointed that the authors did not address the question if the charge-altering mutations of the accessory helix retain the helical character of the accessory helix (suggested by the strong alpha-helical chemical shifts of the accessory helix by*
[38]
*J.Biol.Chem). It is certainly possible that this happens, but the data presented here do not provide direct (e.g., NMR) evidence for this notion. Moreover, the proposed model of electrostatic-steric hindrance by the accessory helix remains speculative. It is of course fine to provide such a speculative model in the Discussion section of the paper. However, the title is misleading since it suggests that the model has been definitely proven in this work. Rather, the main focus of this paper is a re-examination of the interaction between the accessory helix and the truncated SNARE complex that was previously studied by Kummel et al. Thus, we request a change of title to properly reflect the main focus of this work*.

We are sorry that we did not properly address the issue of the potential effects of the charge-altering mutations on the helicity of the accessory helix. Since performing NMR analyses of the mutants would unduly delay publication of the paper, to address the concern related to the potential helix disruption by these mutations we have added the following sentence in the Discussion. However, these results need to be interpreted with caution, since it is plausible that the 5E and 3R mutations may alter the helical character of the accessary helix.

In addition, to address the concern regarding the title of the manuscript, we have changed the title to: 'Re-examining how complexin inhibits neurotransmitter release: SNARE complex insertion or electrostatic hindrance?'

*2) The statement in the Results section: “these cross-peaks actually increased in the spectrum of Cpx(26-83) bound to SCdelta68 with respect to the SC-bound state, and increased somewhat more in the SCdelta62-bound spectrum (*Figure 2—figure supplement 1*) These data show that the synaptobrevin C-terminal truncations increase the flexibility of the accessory helix, in correlation with the destabilization of N terminal part of the central helix,”. The best resolved accessory helix resonances (A30 and Q38) actually do not seem to increase from SCdelta68 to SCdelta62. Thus, the statements about the increase in flexibility of the accessory helix need to be qualified*.

In the original and first revised manuscript, we had adjusted the contour levels of Figure 2 to enable visualization of the weakest cross-peaks of interest. For the expansions showing the cross-peaks of A30 and Q38 in Figure 2, we increased the contour levels proportionately to facilitate observation of the positions of the cross-peaks from the different spectra. Hence, the cross-peak intensities that can be deduced from these plots cannot be directly compared among the spectra. This choice was not meant in any way to bias or confuse the reader, but we realize that it has caused confusion for the review and we are sorry about that. To avoid this confusion in the new revised manuscript and still enable the reader to see the progressive shifts of cross - peaks in Figure 2, without overcrowding the center of the spectra, we have left the same contour levels in Figure 2, but we have adjusted the levels of the expansions of Figure 2 to allow direct comparison of cross-peak intensities. We have explained these choices of contour levels in the figure legend. The expansions of Figure 2 now show that the cross-peaks of A30 and Q38 are considerably stronger for CpxI(26- 83) bound to SC 62 and SC 68 than for CpxI(26-83) bound to SC, as shown quantitatively in Figure 2—figure supplement 1.

*3) The discussion of the Kummel at al. crystal structure (point 4 in the Discussion) remains a concern. Firstly, the area of the interface between the accessory helix and the truncated SNARE complex is actually quite reasonable (715 A^2) and this in itself does not suggest that this contact is induced by “crystal packing”. Secondly, the last statement “that the interactions leading to the zigzag array are not specific and are induced by crystallization” implies that crystal packing may induce non-specific interactions. To the contrary, the observation of such an interaction in different crystal forms and for different mutations of residue 34 (albeit with a register shift observed in the second crystal form for some of the non-crystallographic related molecules) actually speaks against an entirely non-specific interaction. Moreover, the crystallization conditions seem not unusual, and the pH is comparable for the crystallization and NMR experiments. There is one clear difference: in the crystal structure, two complexin molecules are interacting with the truncated SNARE complex. It is theoretically possible that binding of the central complexin helix to the SNARE complex is required in order to enable binding of the accessory helix of the second complexin molecule. Thus, we remain concerned about this difference between the crystal structures and the solution NMR studies on the same system, and the discussion must be modified to properly describe this discrepancy, and provide possible alternative explanations*.

We agree that some of the arguments we used to suggest that the interaction mode observed in the crystals arises from non-specific interactions are not very compelling. Furthermore, it is unclear what can be considered a specific or non-specific interaction in the context of this discussion. Hence, we have removed any mention of non-specific interactions and have discussed the available experimental data on this issue trying to keep an objective perspective.

Among other issues, we now mention that ‘…although the interface area with the SNAREs is larger for the accessory helix (ca. 900 Ă^2^ calculated with PISA; [26]) than for the central helix (ca. 540 Ă^2^ ), the atomic B-factors of the residues of the accessory helix in the interface with the SNAREs are much larger than those in the central helix interface, with little electronic density for the side chains of the accessory helix interface (Figure 1—figure supplement 2). Interestingly, it has been suggested that motion at a crystal packing interface is intermediate between that of a solvent accessible surface and that of a protein core, even for large interfaces (5).’

Since our NMR analyses had focused much more on WT CpxI, but the crystal structure of Kummel et al. was obtained with the superclamp mutant, we have performed additional NMR experiments using the superclamp mutant (supcl). In particular, we have acquired ^1^H-^15^N TROSY-HSQC spectra of ^2^H,^15^N-CpxI(26-83)supcl in the absence and presence of SC 60 (presented in the new Figure 2—figure supplement 2) . For comparison, we have also acquired parallel spectra with WT ^2^H,^15^ N-CpxI(26-83) (shown in Figure 2). As we describe in the manuscript, we obtained very similar results for the WT and supcl proteins, and we were unable to detect an interaction of the accessory helix with SC 60. However, we cannot rule out the possibility that there is a weak interaction in solution that becomes stabilized by crystallization.

*4) The authors may wish to explicitly leave wiggle room for future discovery of some such an interaction, e.g., with a sub-complex of SNAREs. For example, one could write “While current data do not support the proposed zigzag SNARE:complexin interaction, it remains possible that such an interaction might occur under some biological condition, or*
*with some sub-complex of SNAREs.”*

We agree that it is desirable to leave such wiggle room. We believe that such wiggle room is left in the following sentences included in the Discussion; it might be premature to completely rule out these models given the complexity of this system. Note for instance that our NMR and ITC data were obtained with truncated SNARE complexes in solution and hence do not rule out the possibility that the CpxI accessory helix interacts with trans-SNARE complexes partially assembled between membranes. We feel that adding the suggested sentence would be redundant, but would be glad to add it if this required for acceptance of the manuscript.